# The distribution and evolution of supraglacial lakes on the 79 °N Glacier (northeast Greenland) and interannual climatic controls

Jenny V. Turton[1]., Philipp Hochreuther[1]., Nathalie Reimann[1]., Manuel T. Blau[2,3]

[1]Institute of Geography, Friedrich-Alexander University, 90154 Erlangen, Germany
[2]Department of Climate System, Pusan National University, Busan 46241, South Korea
[3]Centre for Climate Physics, Institute for Basic Science, Busan 46241, South Korea

*Correspondence to*: Jenny V. Turton (jenny.turton@fau.de)

**Abstract.** The Nioghalvfjerdsfjorden glacier (also known as 79 North Glacier) drains approximately 8% of the Greenland ice sheet. Supraglacial lakes (SGLs), or surface melt ponds, are a persistent summertime feature, and are thought to drain rapidly to the base of the glacier and influence seasonal ice velocity. However, seasonal development and spatial distribution of SGLs in the northeast of Greenland is poorly understood, leaving a substantial error on the estimate of melt water and its impacts on ice velocity. Using results from an automated detection of melt ponds, atmospheric and surface mass balance modelling and reanalysis products, we investigate the role of specific climatic conditions on melt onset, extent and duration from 2016 to 2019. The summers of 2016 and 2019 were characterised by above average air temperatures, particularly in June, as well as a number of rainfall events, which led to extensive melt ponds to elevations up to 1600 m. Conversely, 2018 was particularly cold, with a large accumulated snowpack, which limited the development of lakes to altitudes less than 800 m. There is evidence of inland expansion and increases in the total area of lakes compared to the early 2000s, as projected by future global warming scenarios.

## 1 Introduction

Nioghalvsfjerdsfjorden, also known as 79° North Glacier (henceforth 79 °N glacier) is a marine-terminating glacier on the northeast coast of Greenland. Approximately 8 % of the Greenland Ice Stream (GIS) drains into 79 °N through the North East Greenland Ice Stream (NEGIS), making it the largest discharger of ice in northern Greenland (Mouignot et al. 2015, Mayer et al. 2018). Prior to the 21st century, NEGIS, which extends 600 km into the interior of the GIS (Figure 1), was believed to be stable, with little change in ice dynamics (Khan et al. 2014, Mayer et al. 2018). However, since 2006 NEGIS has undergone pronounced thinning of 1 m year$^{-1}$, and the floating tongue of 79 °N has retreated by 2-3 km since 2009 (Khan et a 2014). Recently, over 100 km$^2$ of ice was lost through calving of a tributary glacier to 79 °N, Spalte Glacier (Figure S1), following record-breaking summer air temperatures in 2019 and 2020, highlighting the vulnerability of this region to climate change and surface melt.

The surface of 79 °N and the NEGIS feature persistent melt water ponds, or Supraglacial Lakes (SGL), and meltwater drainage channels (Figure S1). SGLs are a frequent summertime feature on many glaciers in Greenland (Pope et al 2016), on ice shelves (e.g Larsen C; Luckman et al 2014) and on sea ice (Perovich et al 2002). The albedo of SGLs is between 0.1 and 0.6, depending on their depth (Malinka et al 2018), and therefore they absorb much more shortwave radiation than the surrounding solid ice (Buzzard et al 2018a). SGLs influence both the Surface Mass Balance (SMB) and the dynamical stability of glaciers through lowering the albedo at the surface and draining water to the base, which reduces friction and influences ice flow velocity (Zwally et al. 2002; Vijay et al. 2019). Both ice velocity increases and decreases have been linked to the drainage of SGLs across Greenland. Short-lived velocity increases have been observed during summer in several marine-terminating glaciers, including 79 °N Glacier (Rathmann et al 2017). Both

Rathmann et al (2017) and Vijay et al (2019) hypothesise that the summer speed-up of 79 °N Glacier occurs when SGLs drain to the base and alter the subglacial hydrology. Conversely, on land-terminating glaciers, SGL drainage has been shown to reduce ice velocity in the seasonal to decadal time scales (Sundal et al. 2011; Tedstone et al. 2015). SGLs are a key component of the SMB and yet rarely feature in mass balance models or estimates (Smith et al. 2017, Yang et al. 2019). Despite the high number of studies focusing on surface mass loss from the Greenland Ice sheet (e.g Lüthje et al. 2006, Das et al. 2008, Tedesco et al. 2012, Stevens et al. 2015), the relationship between SMB, run-off and SGL development remains unclear.

Despite the widespread occurrence of SGLs, very few studies have investigated the relationship between the seasonal evolution of SGLs and the atmospheric processes required for their formation in this region. Previous studies have largely focused on Antarctic ice shelves (Langley et al. 2016, Arthur et al. 2020, Leeson et al. 2020) and southern and western Greenland (Lüthje et al. 2006, Das et al. 2008, Tedesco et al. 2012, Stevens et al. 2015). Recently, more northerly locations have been investigated, including Petermann Glacier (Macdonald et al. 2018). Multispectral satellite products now provide observations of SGL over northeast Greenland at both high-temporal and -spatial resolution, and in many cases free of charge. The northeast of Greenland, and specifically the NEGIS region, has, until recently, lacked such detailed analysis of SGLs, however, this region is likely to show an inland expansion of SGL and ablation zone in the near future (Leeson et al. 2015; Igneczi et al. 2016; Noël et al. 2019). Sundal et al. (2009) used MODIS data to assess the lake area between 2003 and 2007 for 79 °N Glacier amongst other locations. However, as the ASTER images were acquired at a later stage in the melt season, the percentage of unidentified lake area at the start of the summer is likely to be higher than 12 % (Sundal et al. 2009). Winter estimates of liquid water area on the 79 °N Glacier are also now available from Schröder et al. (2020). Recently, Hochreuther et al. (2021) developed an automated melt detection algorithm for Sentinel-2 satellite data. This provides a near-daily, very-high resolution (10m) time series of SGLs on NEGIS during summertime.

Widespread summer melting was observed over Greenland in 2007, 2010 and 2012 due to particularly warm summers and specific teleconnection patterns (Tedesco et al. 2013, Lim et al. 2016, Hanna et al. 2014a). The North Atlantic Oscillation (often termed NAO) is the dominant mode of variability for Greenland and the Arctic, defined as the 'seesaw' of atmospheric surface pressure changes between Iceland and the Azores (Hildebrandsson 1897, Hanna et al 2014b). Three other modes of atmospheric variability were found to be important for specifically the northeast and east of Greenland by Lim et al (2016): the Arctic Oscillation, the East Atlantic pattern, and the Greenland Blocking Index . Generally (for the whole of Greenland), a negative phase of the North Atlantic Oscillation and Arctic Oscillation are associated with a warm and dry atmosphere over the GIS, and often leads to mass loss at the surface (Lim et al 2016). Furthermore, a positive Greenland Blocking Index (especially when combined with a positive East Atlantic pattern and negative North Atlantic Oscillation index) also leads to positive temperature anomalies over the GIS.

More recently, atmospheric rivers, or narrow filament-like regions of intense water vapour transport in the atmosphere, have been investigated in response to extreme surface mass balance variations in the northwest of Greenland (Bonne et al. 2015; Mattingly et al. 2018; 2020). In most cases, the northeast of Greenland, especially the coastal regions and marine-terminating glaciers, have received little or no attention during these stand-out years, possibly due to weaker teleconnection signals (Lim et al. 2016) or due to low spatial resolution data (Oltmanns et al. 2019). Similarly, prior to the mid 2010's, the majority of melting was located in the southern and western parts of Greenland, leading to vast research for these regions (e.g van de Wal et al. 2005; 2012; Tedstone et al. 2017; Kuipers Munneke et al. 2018). However, after the mid 2010's, the highest melt anomalies were located in northern Greenland, especially in 2014 and 2016 (Tedesco et al. 2016). Recently, a low-permeability ice slab was identified in northeast Greenland and within 79 °N Glacier (MacFerrin et al. 2019). The meters-thick, englacial layers of refrozen melt water

enhance melting and runoff processes and are sustained with relatively small amounts of melt water from drainage of
SGLs (MacFerrin et al. 2019). With a warming climate, it is likely that the ice slabs will become more widespread and
persistent, although more research is required to investigate the glacio-hydrology in these regions. In a recent review
paper, Flowers (2018) highlighted that further investigation into surface melt water volume, drainage and runoff from
marine-terminating glaciers is required.
The specific aims of this study are to investigate: 1) the spatial distribution of SGLs over the 79 °N glacier, 2)
the life-cycle of lake development, 3) the atmospheric and topographic controls on melt pond evolution in the northeast
of Greenland between 2016 and 2019 and 4) whether and how conditions have changed since the Sundal et al. (2009)
study in the early 2000s. To accomplish this, we use a combination of very high-resolution (10m) Sentinel-2 data, high-
resolution (1km) atmospheric modelling output from the Polar Weather Research and Forecasting (PWRF) model and
surface mass balance estimates from the COSIPY model, as well as in-situ observations.
In Section 2, we introduce the automatic detection algorithm and data used in the study, followed by the results
(Section 3). These are separated into topographic (Section 3.2) and climatic (Section 3.3) controls of the SGL formation
and spatial distribution. The discussion continues in Section 4 and the research concludes in Section 5.

**2 Data and Methods**
**2.1 Automated SGL detection algorithm**
Automatic SGL detection algorithms have previously been applied to a number of satellite records including MODIS
(Sundal et al. 2009), Landsat8 (Williamson et al. 2018), Sentinel-1 (Schröder et al. 2020) and Sentinel-2 (Williamson et
al. 2018; Hochreuther et al. 2021). A previously developed SGL detection algorithm by Hochreuther et al (2021) has
been applied to Sentinel-2 data between March and September 2016-2019 for melt pond tracking. For a full description
of the processes involved in SGL detection, see Hochreuther et al. (2021), however a brief overview is provided here.
High-resolution (10-60m) optical imagery is collected from two twin satellites, Sentinel-2 A and B, at a revisit duration
of approximately 1-2 days at this latitude. Whilst launched in 2015, data coverage was too low over the study area to
extract a meaningful timeseries of SGLs. Therefore, the timeseries used here runs from March 29 2016 to September 19
2019.

An empirically developed and locally tuned static band ratio threshold for the blue to red band spectra was
applied. This approach was chosen over the often-applied NDWI due to faster computation and expected similar results
(Williamson et al. 2017; Hochreuther et al. 2021). To delineate ice and slush from liquid water, thresholds between 1.0
and 2.4 were tested and compared visually to true colour images, resulting in a best fit at a ratio of 1.6. After the
application of the threshold, the images were cropped to the grounded ice. The GIMP land classification map (Howat et
al. 2014), updated by a Sentinel 2-image from 2016 and combined with an ERS-2 SAR-based grounding line estimation
was used to delineate the eastern ice margin (Hochreuther et al. 2021). Sieving the binary mask, again with iterative size
testing in advance,  reduced noise stemming from crevasse- and serac fields, retaining only water areas larger than 150
pixels (0.015 km2). This potentially causes a number of very small lakes being missed, but represents the best possible
compromise between falsely removing small lakes and falsely retaining misclassifications due to shadows or slush. A
topographic shadow mask was applied to the data to avoid misclassifications. Furthermore, as lakes on the Greenland
Ice sheet have been shown to form mainly within topographic sinks, only water areas within topographic depressions
were retained using a Digital Elevation Model (DEM)-based sink mask, reducing the risk of identifying streams as
lakes. Finally, a two-step cloud detection was applied, taking changes of lake area over time (step 1) and cloud
(shadow) size into account. Depth and volume were not estimated, as no measurements of lake depths exist for similar
latitudes (and thus solar zenith angles) within the observation period of Sentinel-2. Additionally, lakes on 79 °N Glacier
have been shown to partially be significantly deeper than in West Greenland (see Neckel et al. (2020) and discussion
section). As a consequence, spectrum-depth-equations derived in other studies could not be applied here.

Lakes are not automatically detected on the floating tongue portion of the glacier. Firstly, there are no

topographic sinks, as these are reliant on a DEM of the grounded ice sheet. Secondly, the tongue is fast moving
(approximately 1500 m a$^{-1}$; Krieger et al. 2020), which makes it difficult to track the lake outlines from one year to the
next. Finally, melt water on the tongue is extensive and flows in more linear patterns as it drains through crevasses
(Figure S1). Description of the SGLs on the floating tongue throughout the paper reflect only visual inspection of the
satellite images.

**2.2 In Situ Observations**
Observational data at two AWSs located on Kronprins Christian Land (KPC) in the northeast of Greenland are used
from the PROMICE (Programme for Monitoring of the Greenland Ice Sheet) network (https://www.promice.dk, last
accessed 3 April 2019), operated by the Greenland and Denmark Geological Survey (GEUS). AWS KPC_U (Upper) is
located at 79.83 °N, 25.17 °W, 870 m a.s.l and KPC_L (lower) is located at 79.91 °N, 24.08 °W, 370 m a.s.l (Figure 1).
See Table 1 and Turton et al (2019) for more information on data availability and the climatology of this region.

**Table 1: Location, elevation and data availability of KPC_L and KPC_U AWSs. Observations are taken approximately 2m**
**about the surface. T is air temperature, SW$_{in}$ and LW$_{in}$ are the incoming (downward) short and longwave radiation**
**respectively and TSK is the skin temperature of the glacier. See van As and Fausto (2011) for more information on**
**observations from the PROMICE network.**

| Name | Location | Elevation (m a.s.l) | Data Availability | Variables used in this study |
|---|---|---|---|---|
| KPC_L | 79.91 °N, 24.08 °W | 380 | 01.01.2009- present | T, cloud cover; TSK SW$_{in}$, LW$_{in}$ |
| KPC_U | 79.83 °N, 25.17 °W | 870 | 01.01.2009-14.01.2010, 18.07.2012-present | T, cloud cover; TSK SW$_{in}$, LW$_{in}$ |


**2.3 Reanalysis data**
The European Centre for Medium range Weather Forecasts (ECMWF) 5$^{th}$ generation reanalysis product ERA5 has been
developed to replace the ERA-Interim product. ERA5 was gradually released starting in July 2017, and back to 1979 is
now available. The horizontal resolution of ERA5 is approximately 31km and has 137 levels in the vertical from the
surface to a height of 0.01hPa. Total precipitation and snowfall have been extracted from ERA5 at hourly intervals from
the nearest grid point to the coordinates of the AWS. The ratio of snowfall to total precipitation (SF/TP) is then
calculated. Total precipitation and snowfall estimates from ERA5 were compared to observations taken from buoy
measurements in the Arctic Ocean by Wang et al (2019) and found to have a high degree of agreement with
observations. The high resolution of ERA5 was also desirable compared to other available reanalysis products in the
region (Turton et al. 2018).

**2.4 Polar Weather Research and Forecasting Model**
Archived model output from the Polar Weather Research and Forecast (PWRF) model (v3.9.1.1) is analysed.
Meteorological variables are available at daily temporal and 1 km spatial resolution from Turton et al. (2019b) at
https://doi.org/10.17605/OSF.IO/53E6Z. PWRF is a polar-optimised version of the WRF model, to better account for
sea ice and snowpack processes (Hines et al 2015). The majority of adjustments in Polar WRF compared to regular
WRF are located in the Noah land surface module. The model output has been previously evaluated against the in-situ
PROMICE weather stations near 79 °N Glacier and can successfully represent a number of near-surface meteorological
variables for both daily mean and sub-daily timescales (Turton et al. 2020). The full description and justification of the
model setup is provided in Turton et al. (2020) and the inner domain location is presented in Figure 1a. Data are
available from October 2013 to December 2018.

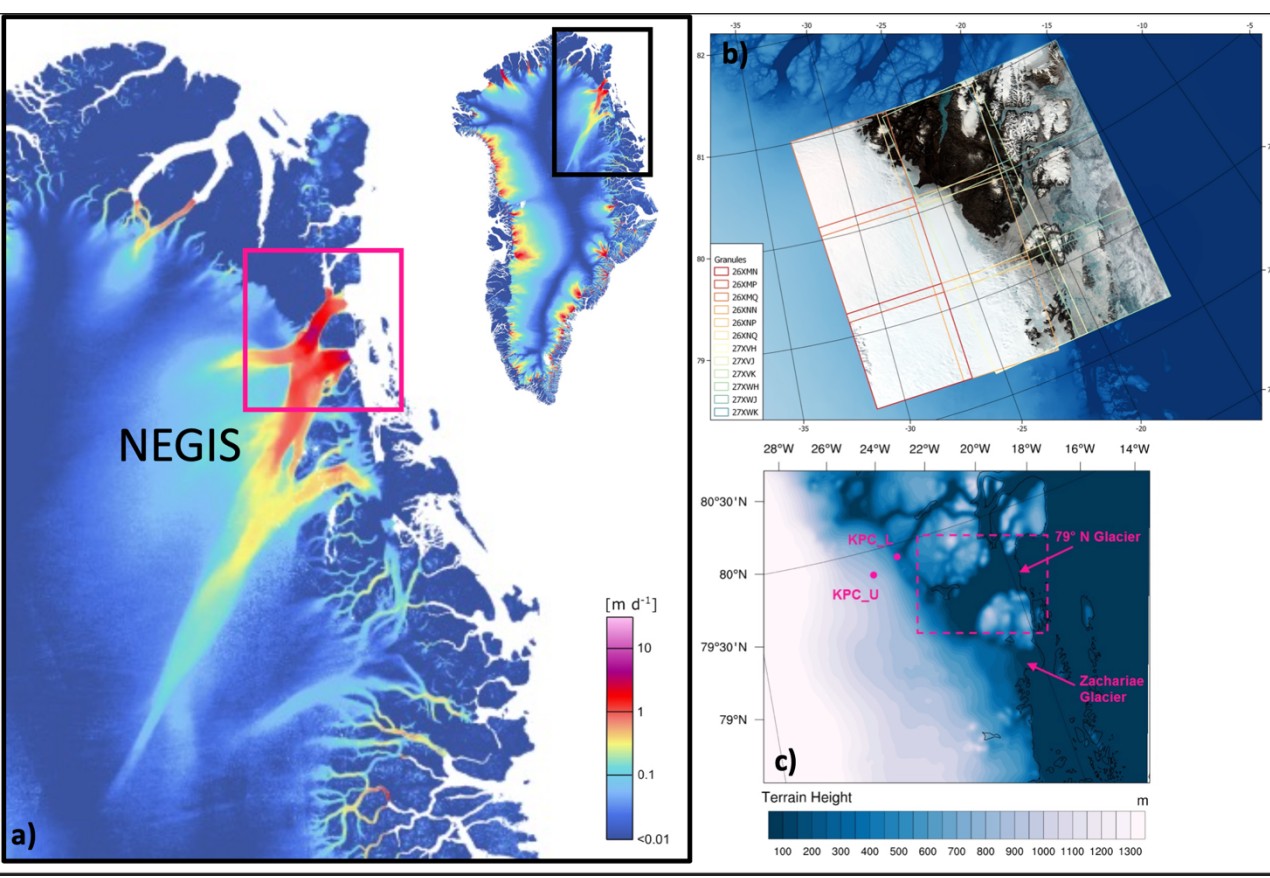

**Figure 1: a) Ice velocity (m d$^{-1}$) of the northeast of Greenland with the North East Greenland Ice Stream (NEGIS) labelled**
**(insert is the whole map of Greenland with ice velocities and a black box outlining the area in a). Pink box outlines the**
**approximate area of b and c. b) The mosaic of Sentinel 2 granules used to apply the SGL detection algorithm, captured on**
**June 19th, 2019. The background is GIMP DEM of Howat et al. (2014). c) The inner domain of Polar Weather Research and**
**Forecasting (PWRF) model simulations by Turton et al. (2020), with the location of the two AWSs (KPC_U and KPC_L) and**
**the elevation of the glacier and ice sheet in colour. The dashed pink box highlights the floating portion of the glacier. Ice**
**velocity data from Sentinel 1, winter campaign from December 2019 to January 2021, from ESA Ice Sheets CCI project**
**(http://products.esa-icesheets-cci.org/products/downloadlist/IV/; last accessed June 20th, 2021).**

**2.5 COSIPY Mass balance model**
To provide an overview of the Surface Mass Balance (SMB) of the region, output from a distributed, open-source SMB
model called COSIPY (COupled Snowpack and Ice surface energy and mass balance model in PYthon)
(https://github.com/cryotools/cosipy; Sauter et al. 2020) is used. Hourly, 1 km spatial resolution surface mass balance

simulations from COSIPY, forced with 4d PWRF output for 2014 to 2018 are used here (COSIPY-WRF). COSIPY-WRF SMB outputs were evaluated against available observations and compared to previous studies by Blau et al. (2021) and found to represent the majority of SMB components with reasonable success at the grounding line and inland for 79 °N Glacier. Archived output from COSIPY-WRF is available at: https://doi.org/10.5281/zenodo.4434259. Here, we use surface mass balance estimates from September 2015 to August 2018 to place our melt pond findings into context of the wider melt in the region. For a full list of parameterisations and description of COSIPY, see Blau et al. (2021).

## 3 Results

### 3.1 Interannual Characteristics

Here, we highlight the important lake characteristics and analyse the climatic and topographic controls responsible for the spatial and temporal distribution of SGLs on 79 °N Glacier, as detected by Hochreuther et al. (2021) from 2016 to 2019. The average size of individual SGLs varies interannually from a maximum of 0.07 km$^2$ in 2016 to 0.02 km$^2$ in 2018.

Typically, lake development began in early June at the lowest elevations. Total lake area increased throughout June and July, reaching a peak in the first week of August. Throughout July, the rate of increase was steady, with approximately 20-25 % increase in lake area from one observation to the next, in all years (Figure 2). From mid-August (day 220-230), the daily change rate became negative as SGLs freeze up or drain. However, in some years there were still individual days of increasing SGL area (positive change rate) punctuating the overall decline in SGL area towards the end of the melt season (Figure 2). This occurred due to periods of warm air temperature or late-season rainfall events. SGLs which remained at the end of the melt season (and have not drained into the firn or channels), typically froze over or became buried in snow. Freeze over of lakes started with a growing floe on one side or with a 'lid' in the centre and freezes outwards (Figure 3). In years with low snow accumulation at the start of September, the frozen, semi-spherical remains of frozen lakes can still be seen.

The rate of increase in SGL area varied interannually (Figure 2). The years 2016 and 2019 were characterised by fast increases in SGL area in June (days 150 to 170-180). In 2016, the increasing rate of SGL area regularly exceeded 100% increase in total SGL area from one observation to the next (Figure 2). June 2017 had a relatively steady increase in SGL area, with approximately 25 % daily increases in area. June 2018 was characterised by a see-saw pattern in expansion of lake area, with periods of fast increases in area (approximately 50% daily increases), followed by periods of SGL lake closure (Figure 2). Sustained expansion of lake area only occurred after the last week of June for 2018. Closure or freeze-over of lakes at the end of the melt season was later and slower in 2018 than in 2016, 2017 and 2019 (Figure 2), and some lakes even remained open at the end of the observation period in mid-September.

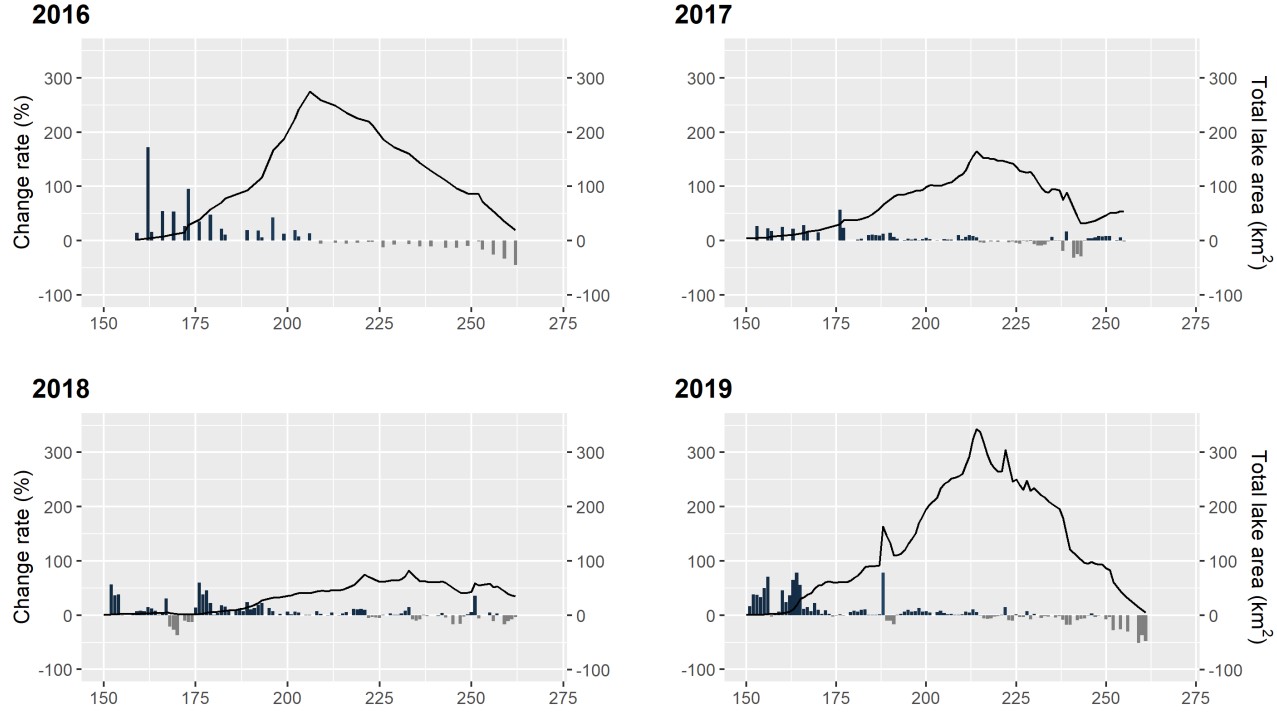

**Figure 2: Change rates of the lake area between observations from 2016 to 2019, limited to DOY 150 – 270 (bars, in percent of the last observed area). Line graph: Total lake area in km².**

Similar to the rate of change, the total SGL area varied interannually. The largest peak total SGL area was seen in 2019, with 330 km² (Figure 2). Conversely, the smallest peak total SGL area was in 2018 with just 77 km² (Hochreuther et al. 2020). This is approximately a 329 % increase between maximum lake area in 2018 and in 2019. The spatial difference in the years is shown in Figure 3, where considerably more lakes are highlighted in 2016 and 2019 than in either 2017 or 2018. Whilst this only shows a snapshot of conditions on two different days, representing peak conditions (mid-July; blue) and a period when the SGLs freeze up (mid-August; pink), the spatial distribution of the lakes differs by years. SGLs at elevations greater than 800m are detected across much of the glacier in 2016 and 2019, but only sparsely in 2017 and 2018 (Figure 3 and 4). Similarly, much larger SGLs are open in 2016 and 2019 than the other two years (Figure 3). The peak total SGL area in 2016 and 2019 was considerably larger than in 2017 and 2018, especially at altitudes from 1000 to 1600 m a.s.l (Figure 4). However, in years with a lower total SGL area, such as 2018, the distribution of lakes is skewed more towards lower elevations (Figure 4c).

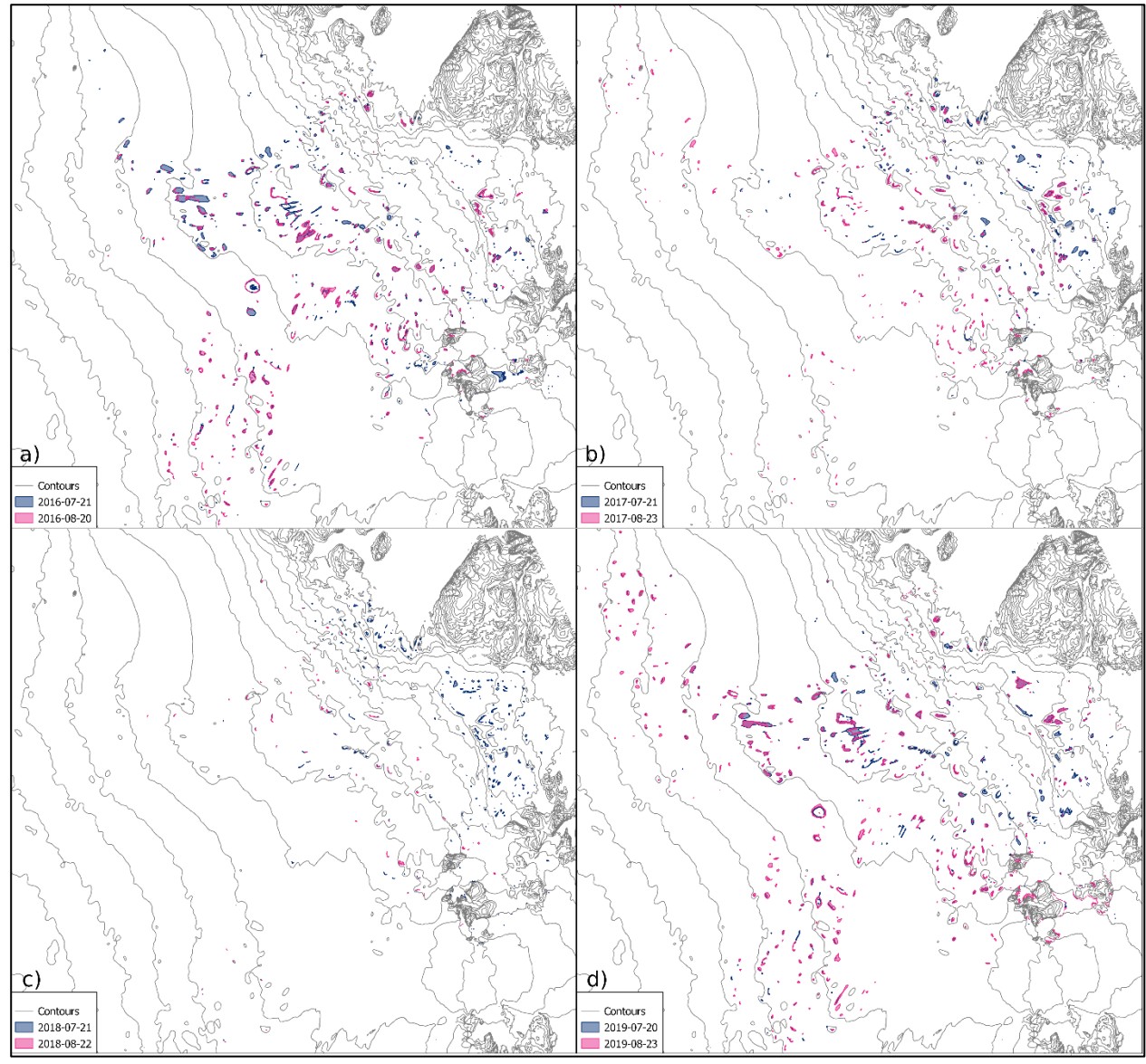

**Figure 3: Lake area in July (blue) and August (purple) for 2016 (a), 2017 (b), 2018 (c) and 2019 (d). Contours are every 100m. Lakes on the tongue have been removed to assess only those controlled by topography.**

**3.2 Topographic Controls**

Melt lakes are part of the whole drainage system of ice sheet hydrology. Given sufficient meltwater availability, the location of lake formation is foremostly controlled by the topography of the ice sheet surface (Lüthje et al 2006). Lakes therefore act as a sink for the englacial channels which distribute the water across and through the ice sheet. The position of lakes on the Greenland Ice Sheet is therefore largely controlled by the underlying bedrock topography (Lampkin and Vanderberg, 2011).

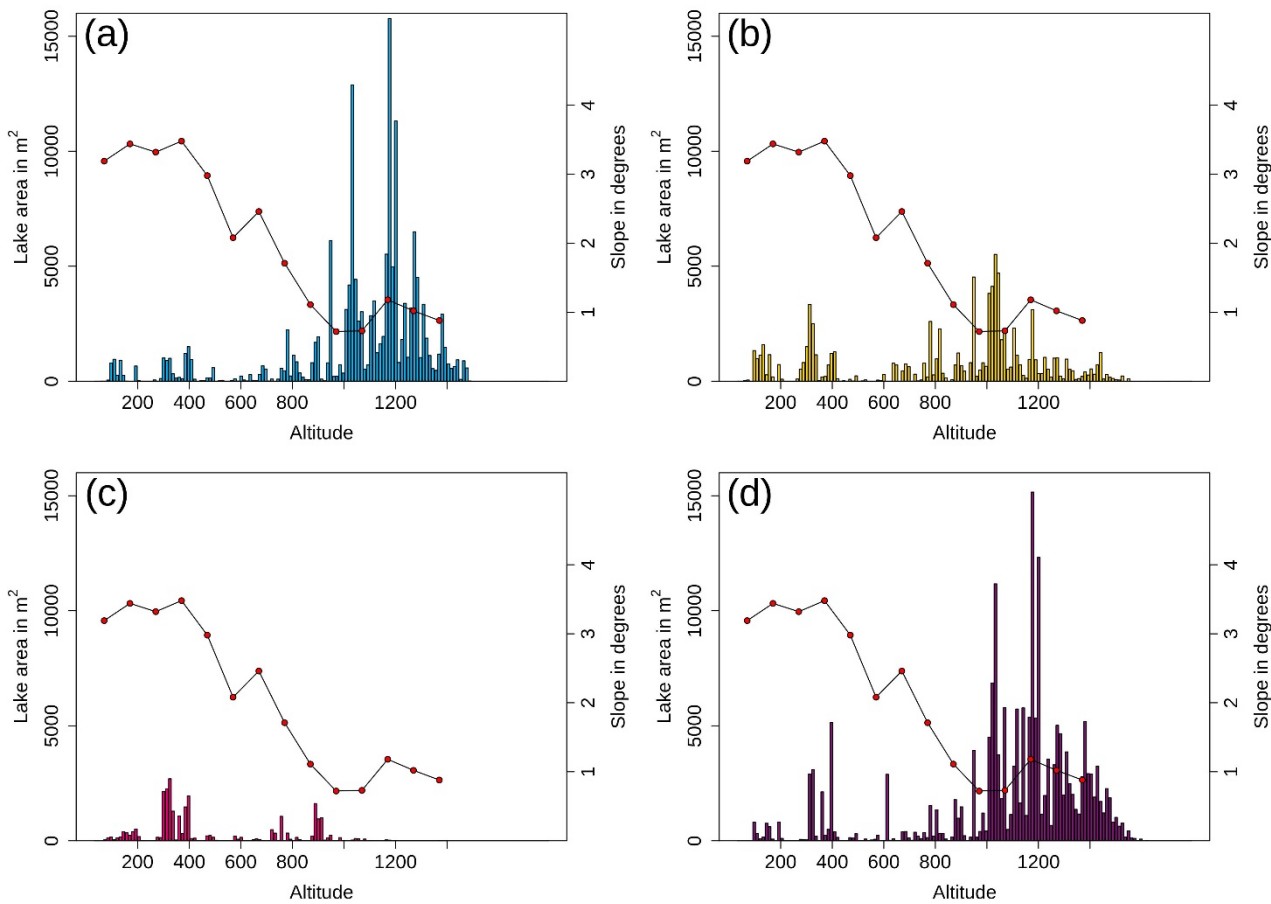

**Figure 4: Altitude distribution of lake area for the maxima of 2016 (a), 2017 (b), 2018 (c) and 2019 (d) per 10m altitude difference. Red dots show average slope angle for 100m altitude bins.**

Below the grounding line of 79 °N Glacier (on the floating tongue), the lakes advect downstream with the flow of the glacier towards the ocean, in a similar fashion to those observed on Petermann Glacier (Macdonald et al. 2018). However, above the grounding line, lakes develop in the same depression or location each year (Figure 3). The SGL area in 2016 and 2019 is larger compared to 2017 and 2018. This interannual change in SGL area is due to the inland expansion of lakes to higher elevations (Figure 3), as opposed to the development of new lakes at lower elevations.

The minimal SGL area between approximately 200 m and 600 m (Figure 4) is partly a consequence of higher slope angle. The slope of the glacier surface between these altitudes is approximately 3 ° to 4 °. The areas with larger SGL area and where the largest lakes develop (Figure 3) is between 0.6 ° and 1.5 ° (Figure 4). Unlike some of the ice shelves in Antarctica, where SGLs are concentrated around the grounding line due to low elevation and slope (Arthur et al. 2020), on 79 °N Glacier, SGLs are also clustered at higher altitudes, where low slope angles are also measured. Consequently, the largest lakes can be found at altitudes between 850m and 1000m. The highest elevation of SGL development was at 1600 m in 2019 (Figure 4). Due to the flat terrain, these lakes are, judging from the blue spectrum saturation, comparatively shallow, whereas the lakes close to the grounding line appear smaller in area but deeper (Figure S2).

Significant decreases of total lake area can be attributed either to sudden climatic changes, or to consecutive drainage events. In 2019, the sudden decrease around DOY 240 is attributed to a large freeze-over of the majority of all lakes above 700 m a.s.l. Conversely, the decrease following the 2019 peak of total lake area on August 2[nd] (DOY 214) was caused by a step-wise drainage pattern, starting with larger lakes at high altitudes, followed by drainage events close to the ice front of Zachariae and accompanied by a speedup of calving and seawater movement (Figure S3).

Because of the timing and sequence of the rapid drainage events, we can deduce a subglacial meltwater reconfiguration
in this case.

**3.3 Climatic Controls**

Whilst the location of the individual lake is controlled by topographic features, whether or not the lake will develop is
due to atmospheric conditions. In conjunction with the topographic controls, the second most important control for lake
development is the availability of melt water, which is largely controlled by the weather conditions. We have assessed
numerous atmospheric variables for the four-year period, in an attempt to investigate the relationship between these
variables and the melt onset and extent.

Buzzard et al. (2018a) investigated the impact of varying atmospheric variables in an idealised 1-D melt pond

model and identified that near-surface air temperature ($T_a$), skin (or surface) temperature (TSK), shortwave incoming
radiation (SWin) and snowfall (SF) had a considerable impact on the development of SGLs. We investigate these
variables in conjunction with rainfall following the findings of Oltmanns et al. (2019). Other previously investigated
variables which had little to no influence on SGL development include wind speed and non-climatic variables such as
wet-snow albedo (Buzzard et al. 2018a), which we do not investigate.

**3.3.1 Air Temperature ($T_a$)**

The average summer (JJA) $T_a$ is 0.7 °C over the floating tongue of the glacier, decreasing to -1.2 °C at an elevation of
830 m a.s.l observed at KPC_U AWS (Turton et al 2019). The average June, July and August air temperatures at
KPC_L (KPC_U) are 1.1 °C (-2.1 °C), 3.6 °C (0.7 °C) and 0.5 °C (-2.6 °C) respectively (see Figure 1 for AWS
locations). Typically (from 2009-2019), the daily average $T_a$ reaches 0 °C in the second week of June at approximately
390 m a.s.l (KPC_L location), and late June at 830 m a.s.l (at KPC_U location) (Table 2). From this date until mid-
August, the daily air temperatures are often at or just above the melting point (Figure 5).

In 2016, all three summer months observed above average $T_a$ at both observation sites. At higher elevations,

daily $T_a$ reached 0 °C slightly earlier than usual (June 11, 2016), after a cooler than average start to June, especially at
KPC_U (Figure 5a). Rather than a gradual increase in air temperatures throughout the start of June, there was a marked
jump in temperature between June 5 and June 11, 2016 (Figure 5a). At KPC_U the temperature increased from -10.1 °C
on June 5 to 0.9 °C on June 11, and then remained above or close to freezing (0 °C +/- 0.75 °C) until mid-August
(Figure 5a). Just 16 days after this temperature jump, SGL formed at elevations of approximately 870 m a.s.l (elevation
of KPC_U) (Table 2; Figure 5a). There were 84 days (70 of which were consecutive) with above-zero daily $T_a$ in 2016
at KPC_L (Table 2). The longest consecutive period with above average air temperatures at both KPC_L and KPC_U,
from observations between 2009 and 2019, was during 2016. The average June 2016 $T_a$, simulated by PWRF, was
above freezing for large parts of the NEGIS region (Figure 6a). Spatially, these higher air temperatures approximately
follow the 800 m contour line, showing some agreement with the altitude-temperature relationship. However, the July
2016 average air temperatures deviate from this relationship, with warmer air temperatures above 1200 m for the 79 °N
Glacier but remaining below 800m near Zachariae and to the south of the glacier (Figure 7a). Average July 2016 $T_a$
above 3 °C is simulated for large parts of NEGIS. At KPC_L, July 2016 was 3.2 °C warmer than average, agreeing well
with the PWRF data.

The earliest observation of above-zero daily $T_a$ (from 2009 to 2019) was May 27, 2017 at KPC_L. However,

air temperatures rapidly decreased again at the end of May, 2017, before reaching 0 °C on June 1, 2017 (Figure 5b).
Both June and August 2017 average air temperatures at both observation sites were slightly below average, but the July
average temperature was 0.5 °C (0.2 °C) warmer than the 2009-2019 average at KPC_L (KPC_U). Despite the lower

June Ta in 2017 compared to 2016, the length of time between Ta reaching above 0 °C at KPC_L and development of melt ponds at 370 m a.s.l. was also 14 days (Table 2; Figure 5b). However, at higher altitudes, there were only 5 days between Ta above 0 °C and melt ponds developing at 870 m a.s.l. in 2017 (Table 2; Figure 5b). The cooler air temperature in 2017 relative to the previous summer is evident over the majority of NEGIS, with above average Ta locations restricted to low elevation pockets (Figure 6b). The average 2017 Ta is spatially more similar to the 2016 situation in July (Figure 7). In July 2017, Ta greater than 0 °C was simulated over much of the 79 °N Glacier, up to elevations greater than 1000 m a.s.l. Lower elevation regions, and areas of seasonally exposed rocks reached daily average Ta of 3 °C (Figure 7b). At higher elevations, the earliest closure of SGLs within the four-year period was observed in this year (September 1, 2017 at 870 m a.s.l), which was approximately 10 days after the Ta dropped below freezing at KCP_U (Figure 5b). Similarly, at lower elevations, 2017 saw the earliest SGL closure of the four-year period on September 12, 18 days after Ta dropped below freezing at KPC_L (Table 2; Figure 5b).

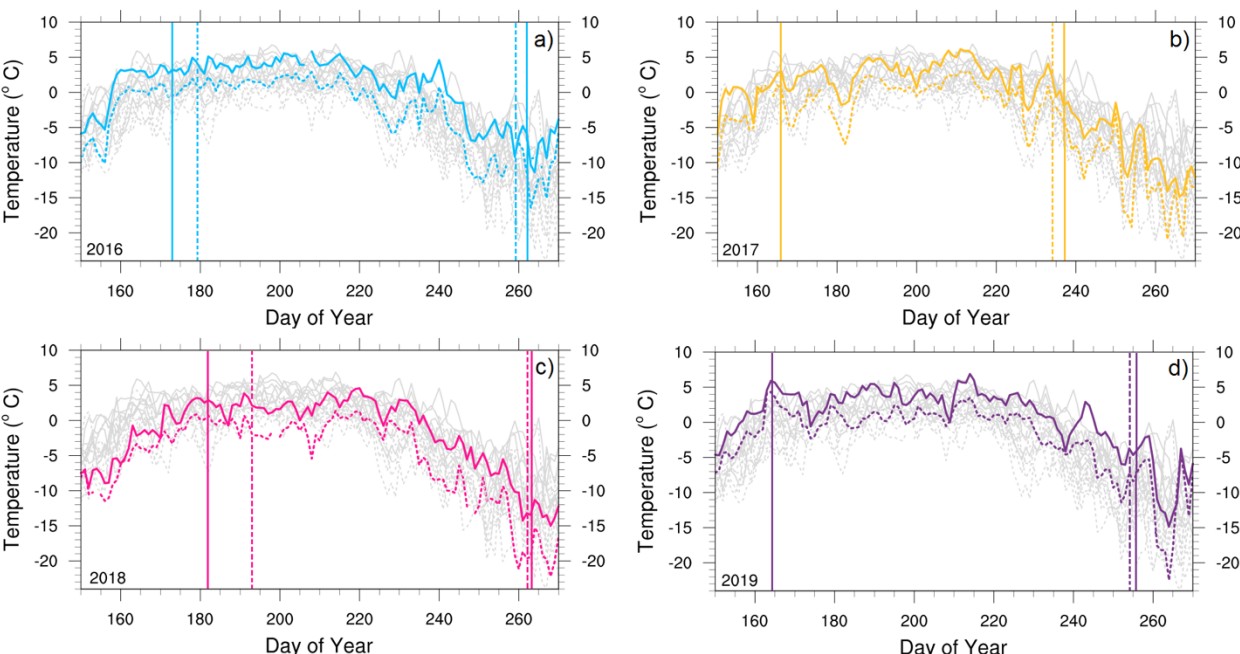

**Figure 5: The daily air temperature observations from KPC_L (solid line) and KPC_U (dashed line) from day of the year 150 to 270 for a) 2016, b) 2017, c) 2018 and d) 2019. Grey lines are daily air temperature from 2009-2019 (when available). Vertical solid (dashed) lines represent the opening and closing of SGLs at KPC_L (KPC_U) approximate elevations (information from Table 2).**

The smallest total SGL area and latest lake development were observed in 2018. The latest observed onset of warm air temperatures was also in 2018, when the first recorded above-zero daily Ta was on June 20, 2018 (Figure 5c; Table 2). This is also evident in the later onset of SGLs at both lower and higher elevations (Figure 5c; Table 2). The first two weeks of June 2018 were colder than in any other year in the last decade of observations (Figure 5c). This is also reflected in the much colder June average Ta over the NEGIS region from the PWRF 2018 simulations (Figure 6c). All three summer months in 2018 were characterised by considerably cooler air temperatures over the area of interest, with above freezing temperatures restricted to very low-lying parts of the glacier during July (Figure 7). June and July 2018 were both 2.0 °C cooler than average at both observation locations. Both the number of days above freezing and the consecutive number of days above freezing were both at their lowest in 2018 (Table 2), with just 8 consecutive days

above freezing at KPC_U. In August 2018, Ta increased and were close to average conditions throughout August (Figure 5c). The last day with Ta above freezing was observed on August 25, 2018 at KPC_L, the same as in 2017 (Table 2). However, the latest observation of SGLs at 370 m a.s.l was September 18, 2018, the latest in the four-year period, and SGLs were still visible at the end of the observational period (Table 2; Figure 5c).

At lower elevations, the conditions in summer 2019 were remarkable. At both KPC_L and KPC_U, air temperature records were broken in June 2019 (Figure 5d), along with most areas of the ice sheet (Tedesco and Fettweis, 2020). There were 115 days of Ta greater than 0 °C with 61 of those being consecutively observed at KPC_L in 2019 (Table 2). Similarly, warm Ta continued past the summer season, with the final observation of Ta above 0 °C on September 28, 2019 (Table 2). On June 12, 2019, a new daily air temperature record was set at KPC_U of 4.2 °C, swiftly broken by a new daily record on June 13 of 4.3 °C. Prior to these two days, the highest temperature had been during the record-breaking summer of July 2012. Similarly, an hourly maximum of 7.9 °C was recorded at KPC_U, which is the highest hourly temperature observation in a decade. Despite a warm start to the season, air temperatures returned to normal for the remainder of June and July. A second peak temperature event was recorded in early August 2019. The highest daily air temperature record at KPC_L (between 2009 and 2019) of 6.9 °C was observed on August 2, 2019. The spatial distribution of the Ta in summer 2019 is not analysed as PWRF simulations are not available for this period. However, satellite images reveal extensive surface melt pond formation, very thin and broken sea ice, and a 50 km$^2$ calving event of Spalte Glacier was also recorded this year (Figure S1). When taken altogether, these characteristics point to particularly warm temperatures across the whole region in 2019. SGL development started earlier in 2019 than in 2016 despite both years observing Ta above 0°C at a similar time (June 6 in 2019 and June 7 in 2016) (Table 2; Figure 5a,d).

**3.3.2 Skin Temperature (TSK)**

When daily average TSK is at 0 °C, the term TSK$_{melt}$ is used in this manuscript to represent likely surface melting. At KPC_L, the average (2009-2019) melt day onset is June 18, whereas at KPC_U this date is June 28. The average number of days with TSK$_{melt}$ is 44 at KPC_L and 12 at KPC_U. The average number of consecutive TSK$_{melt}$ days is 21 at KPC_L and 5 at KPC_U.

In terms of the skin temperature of the glacier at KPC_L location, 2016 stands out. The largest number of TSK$_{melt}$ days and longest number of consecutive TSK$_{melt}$ days were observed in 2016 (64 days, of which 47 were consecutive). Similarly, the earliest onset of TSK$_{melt}$ in the four-year period was observed at KPC_L in 2016, on June 9. At KPC_U, the number of TSK$_{melt}$ days and consecutive TSK$_{melt}$ days were also above average for 2016, however the onset of surface melt was later than usual (July 1). Not only was this a stand-out year at KPC_L from the four-year study period, but also in the observational record from 2009. Even the record-breaking melt year of 2012 had fewer TSK$_{melt}$ days and consecutive melt days.

The year 2017 was a relatively average melt season in terms of TSK$_{melt}$. The onset of TSK$_{melt}$ at KPC_L was on June 13 (only 5 days earlier than average) and there were 46 TSK$_{melt}$ days, of which, 17 were consecutive. At KPC_U, the melt onset was earlier than average (June 10) but the number of TSK$_{melt}$ days and consecutive melt days were lower than average (9 and 3 days respectively). The latest melt onset date was observed in 2018 at both locations; June 26 at KPC_L (8 days later than average) and August 3 at KPC_U (36 days later than average). At KPC_U, only one day observed TSK$_{melt}$ and only 30 days (13 consecutive) experienced TSK$_{melt}$ at KPC_L. Therefore, the shortest melt duration and latest melt onset at both locations were observed in 2018.

The year 2019 has a distinct spatial characteristic in terms of TSK$_{melt}$. At lower elevations, the number of TSK$_{melt}$ and consecutive TSK$_{melt}$ days are below average (27 and 17 respectively). However, at higher elevations,

melting is above average with 17 TSK_melt days, of which 6 were consecutive. At KPC_U, TSK_melt onset was also earlier
than average. Despite the above average Ta at PKC_L and KPC_U in 2019, only above average TSK conditions were
observed at KPC_U.

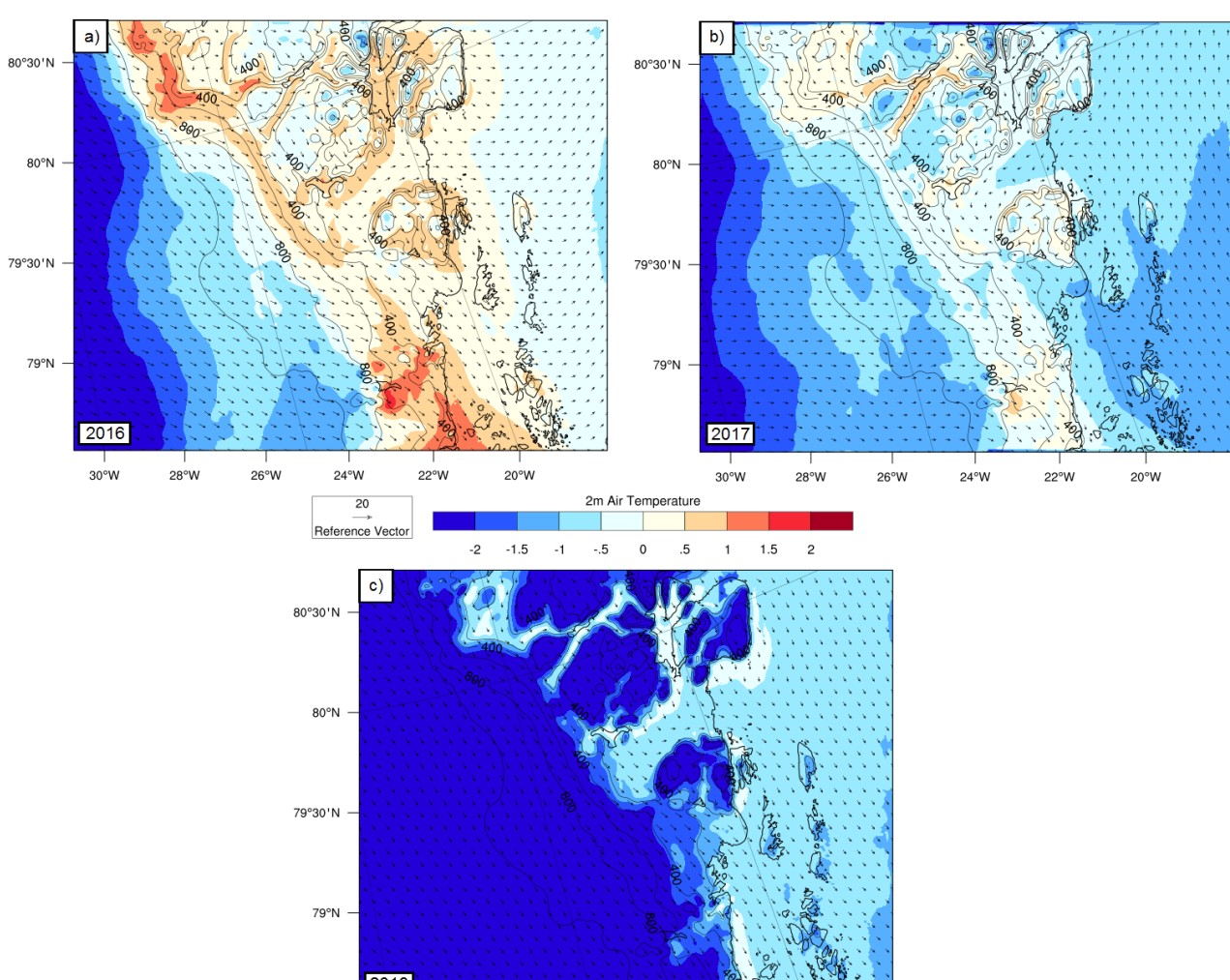

**Figure 6: The monthly average 2m air temperature from PWRF runs for June 2016 (a), 2017 (b) and 2018 (c). Simulations**
**were not available for 2019. Contours are every 200 m, with labels every 400 m. Black arrows are wind vectors displaying**
**wind direction and speed, with a reference vector of 20 ms$^{-1}$ provided.**

**3.3.3 Incoming Shortwave Radiation (SWin)**
In 2016, June and July both experienced positive biases in SWin at both observation sites. At KPC_L, the SWin was 7.3
Wm$^{-2}$ and 16.7 Wm$^{-2}$ higher than average for June and July (respectively). At KPC_U, a positive bias of 10.2 Wm$^{-2}$
during June and 6.4 Wm$^{-2}$ in July was observed in 2016. There was also a positive bias of 17.3 Wm$^{-2}$ and 7.5 Wm$^{-2}$
observed in July 2017 (KPC_L and KPC_U respectively). This increase in SWin observed at the surface is attributed to
less cloud cover in the region. Cloud cover (fraction) at the KPC stations is estimated from downwelling longwave
radiation and air temperature (both of which are observed) (Van as 2011). There was a reduction in cloud cover fraction
in June, July and August in 2016 at both locations. The average summer cloud cover fraction at both locations is 0.4,
whereas in 2016 it was 0.3. The reduced cloud cover is further evident in the sentinel images, with many more clear-sky
days over NEGIS in 2016 than 2017 or 2018.
The SWin was lower than average at both observation sites in June 2017 (-2.6 Wm$^{-2}$ at KPC_U and -10.5 Wm$^{-2}$
at KPC_L). There was a positive bias in SWin of 17.3 Wm$^{-2}$ and 7.5 Wm$^{-2}$ observed in July 2017 (KPC_L and
KPC_U respectively), revealing clear skies in July. At lower elevations, this positive bias continued into August, with a
monthly average bias of 6.7 Wm$^{-2}$ at KPC_L. However, at KPC_U, a negative bias of -8.5 Wm$^{-2}$ was observed.
Despite the cooler conditions at both locations in summer 2018, positive biases in SWin were observed at both
locations in July and August. The July SWin average was 32.7 Wm$^{-2}$ and 18.4 Wm$^{-2}$ higher than the 2009-2019 average
at KPC_L and KPC_U, respectively. Similarly, the August SWin positive bias was 18.9 Wm$^{-2}$ at KPC_L and 17.3 Wm$^{-2}$
at KPC_U. Higher than average cloud cover in June (0.45 compared to 0.36 at KPC_U) and lower than average in July
and August provide further evidence for clearer skies in the mid to late summer. The positive SWin and average
temperatures towards the end of summer 2018, together with a considerable amount of liquid water from the melted
snowpack, likely provided optimal conditions for the later peak in maximum SGL area and slower freeze over of the
lakes, with many still remaining open at the end of the observational period in September 2018 (Table 2).
Some of the largest anomalies of SWin were observed in summer 2019, with KPC_L and KPC_U observing
monthly negative anomalies of -30.0 Wm$^{-2}$ and -19 Wm$^{-2}$ respectively, for June, despite the high temperatures.
Conversely, July saw opposite anomalies, with large positive anomalies in SWin at both KPC_L (+35.4 Wm$^{-2}$) and
KPC_U (+34.3 Wm$^{-2}$). Similarly, the July average cloud cover was considerably below average, with a value of 0.24
compared to an average of 0.36 at KPC_U. A persistent high-pressure system was responsible for the early-season
temperature and melt increases seen over the whole ice sheet (Tedesco and Fettweis, 2020). However, increased
cloudiness observed in the northeast of the ice sheet (and also simulated by Tedesco and Fettweis, 2020) also
contributed to the early melt onset in June.

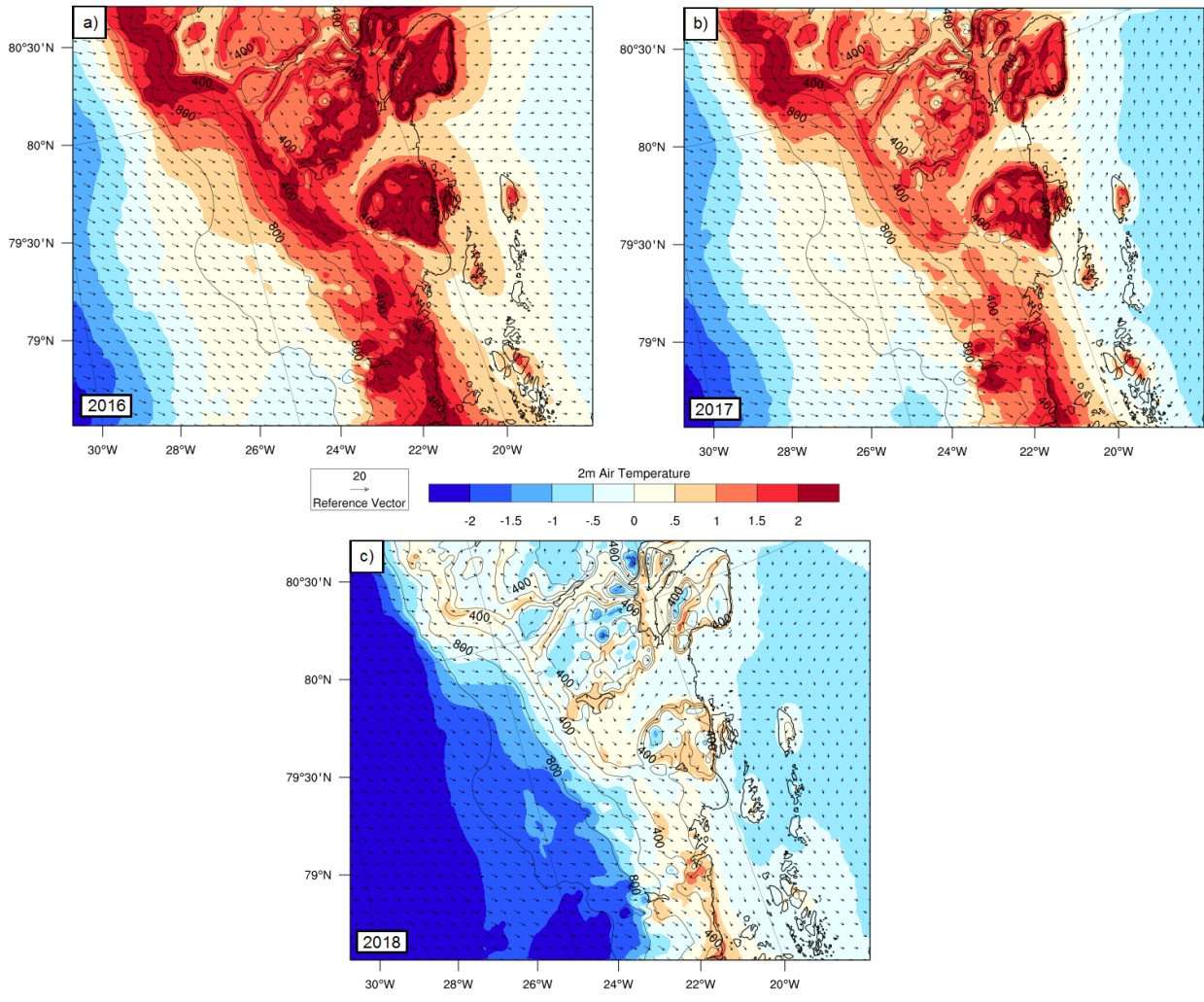


**Figure 7: The monthly average 2m air temperature from PWRF runs for July 2016 (a), 2017 (b) and 2018 (c). Simulations were not available for 2019. Contours are every 200 m, with labels every 400 m. Black arrows are wind vectors, displaying wind direction and speed, with a reference vector of 20 ms⁻¹ provided.**



**Table 2: The timing of the first (last) daily average Ta greater than 0 °C (Ta > 0 °C), number of days with daily Ta greater than 0 °C and earliest development (freeze up) of melt ponds at elevations closest to the AWS elevations. 370 m a.s.l. relates to KPC_L elevation and 870 m a.s.l. relates to KPC_U elevation. *One day observed just below 0 °C in this period. ** end of sensing period. Melt pond development and freeze over dates are represented in Figure 5.**

| Year | AWS | Ta > 0 °C | # days Ta > 0 °C (consecutive) | SGL develop at 370 m/870 m elevation | Ta consistently < 0 °C | SGL freeze over at 370 m/870 m elevation |
|------|-------|----------|-------------------------------|--------------------------------------|------------------------|------------------------------------------|
| 2016 | KPC_L | June 7 | 84 (70*) | June 21 | Aug 30 | Sep 18** |
|      | KPC_U | June 11 | 79 (44) | June 27 | Aug 29 | Sep 15 |

| 2017 | KPC_L | June 1 | 85 (39) | June 15 | Aug 25 | Sep 12 |
| | KPC_U | June 10 | 73 (16) | June 15 | Aug 22 | Sep 1 |
| 2018 | KPC_L | June 20 | 66 (38) | July 1 | Aug 25 | Sep 20** |
| | KPC_U | June 26 | 51 (8) | July 12 | Aug 16 | Sep 19 |
| 2019 | KPC_L | June 6 | 115 (61) | June 13 | Sept 29 | Sep 13 |
| | KPC_U | June 12 | 67 (14) | June 13 | Aug 18 | Sep 11 |

### 3.3.4 Total Precipitation (TP) and Snowfall (SF)

As precipitation is not observed at the KPC stations, we have used ERA5 data. Following Wang et al (2019), a high ratio of snowfall to total precipitation can be inferred as more snow, whereas a low ratio means more precipitation fell as rain than snow. Between September 2015 and May 2016 (accumulation period), 160mm of cumulative snowfall fell at the KPC_U location. The ratio of snowfall to total precipitation was 1.0, meaning that all precipitation fell as snow. However, during summer 2016, especially July and August, some rainfall is present in the region (Figure 8). In July 2016, all 7.7 mm of cumulated precipitation was liquid rain (ratio of 0), and in August, the ratio was 0.82 with 1.9 mm of rainfall. For the whole summer period (JJA), the ratio was 0.5. Even though the summer was therefore relatively dry, there was still a larger amount of summer rainfall in 2016 than in other years.

Total accumulated snowfall between September 2016 and May 2017 at KPC_U was approximately 130 mm w.e, which is the second lowest total amount in our four-year period of interest (Figure 8). The summer (JJA) 2017 snowfall to total precipitation ratio was 0.96, highlighting the minimal rainfall in this year: the smallest rainfall total in the four-year period.

The largest amount of cumulated snowfall during the accumulation period (September to May) occurred in 2018 with 277.9 mm (Figure 8). In the other years of interest, the cumulated snowfall total was less than 190 mm. There were a number of large snowfall events in 2018 which contributed to the larger total precipitation. For example, between February 22 and February 26, 2018, 56.5 mm w.e snowfall fell in the region, which is more than the winter (DJF) total snowfall in 2015/2016. The regular fresh snow episodes increased the albedo and reflected shortwave incoming radiation at the start of the summer season. A thick, fresh snowpack also has a low density, with more space for liquid water to penetrate instead of sitting on the surface in SGLs. The switch from SGL area increase (lake development) to decrease (freeze up) and back again during June 2018 (Figure 2) was due to a number of snowfall events in June, which covered any exposed SGLs. The continuous input of snowfall throughout the year and into summer delayed the onset of SGL development at 870 m a.s.l to mid-July 2018 (Table 2), which was the latest in the four-year period.

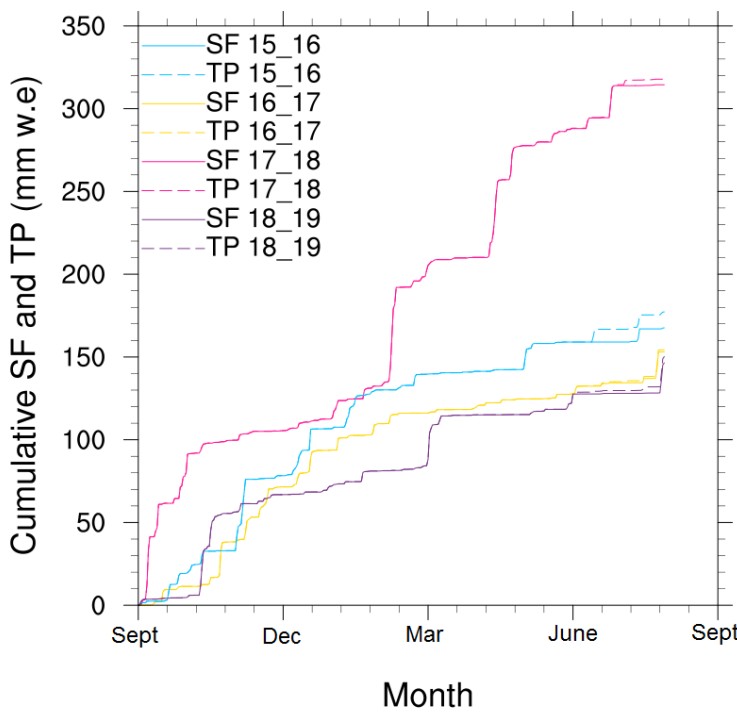

**Figure 8: The cumulative total precipitation (TP) and snowfall (SF) from September (beginning of the accumulation season) to August (end of melt season) at KPC_L location from ERA5.**

The smallest accumulated snowfall from 2016 to 2019 occurred in 2019, with only 125 mm falling by May 2019 (Figure 7). The particularly shallow snowpack provides less water storage availability and lower albedo values, which likely led to the earlier SGL detection in 2019 compared to the other warmer than average year of 2016. The later refreeze of SGLs in the previous summer may also have contributed to the earlier detection in 2019. At the end of August 2019, 21 mm of snowfall occurred, which started the new accumulation season earlier than in previous years (Figure 8). Visual analysis of Sentinel 2 data reveals that between August 30 and September 16th, 2019 there were very few melt ponds detected due to thick cloud cover. On September 20, 2019, there is evidence of fresh snowfall and very few pond outlines remaining, which agrees with the ERA5 analysis of snowfall towards the end of August and start of September.

### 3.3.5 Climate Influence Summary

Summer 2016 experienced the largest average individual SGL size (0.07 km$^2$), second largest total SGL area and second fastest rate of SGL area growth in our four-year record. A combination of above average air temperatures, particularly in mid-June and July, and a large amount of liquid precipitation during summer was likely responsible for the rapid SGL development and peak in total SGL area in late July. Despite the early observation of Ta above freezing in 2017, the earliest in our four-year period, the June 2017 average Ta was slightly below average. This, combined with the slightly above average July 2017 temperatures, likely led to the slower rate of increase in SGL area in 2017 compared to 2016 (Figure 2), and peak in maximum area in early August 2017. The thinner snowpack and limited amount of liquid precipitation falling during summer contributed to the lower maximum SGL area of 153.26 km$^2$ in 2017, compared to 265.39 km$^2$ in 2016. In 2018, the spatial distribution of SGLs was different to the other three years, with the largest SGL area at elevations between 300 m and 400 m a.s.l (Figure 4). Very few SGLs were observed at elevations greater than 900 m, leading to smaller average individual SGL area, as no larger lakes at higher elevations

were identified (Figure 3). Average individual lake size in 2018 was 0.02 km$^2$, compared to 0.07 km$^2$ in 2016, 0.06 km$^2$
in 2017 and 2019. A combination of the cooler air temperatures at the start of summer (see Section 3.3.1) and thick
snowpack led to the delayed onset of SGL development, lower maximum altitude of SGLs and lower total SGL area in
2018 (Figure 3). Total SGL area was largest in 2019, even though the average size of individual SGLs was the same as
in 2017 (0.06 km$^2$). A combination of higher air temperatures, more days above freezing and a smaller snowpack at the
start of the melt season all contributed to a significantly higher total SGL area in 2019 (Figure 4). The peak melt pond
area at the start of August 2019 coincides with an air temperature peak of 6.9 °C on August 2$^{nd}$ at KPC_L, the warmest
daily Ta ever recorded here (Figure 5).
To summarise the climatic conditions: We find that a combination of above average air temperatures, a thin
pre-summer snowpack and summer precipitation falling as rain during summer 2016 and 2019 led to the exposure of a
large number of SGL over a much larger area than observed in the two other years. Conversely, a large amount of
snowfall preceding the melt season and below average air temperatures in 2018 led to the development of very few
SGLs, which were restricted to the lower elevation areas.

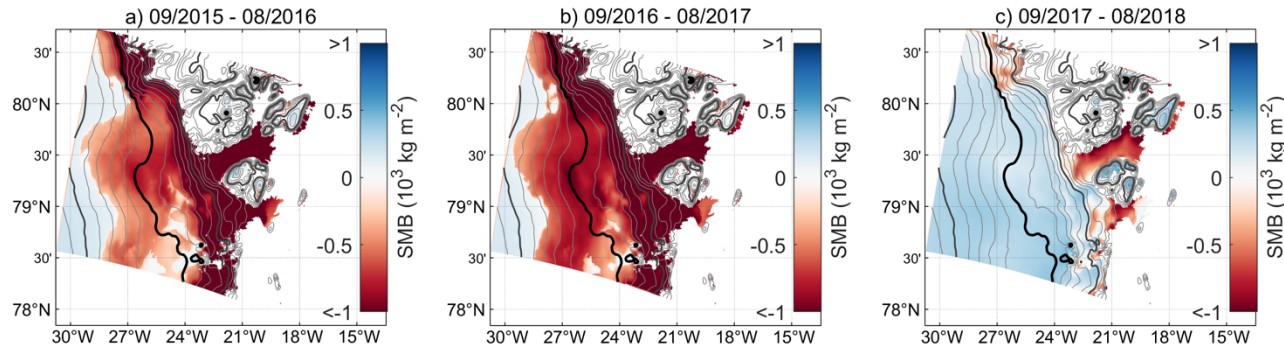

**Figure 9: The annual surface mass balance of the 79°N glacier and NEGIS region from September to the following August in**
**2015-2016 (a), 2016-2017 (b), 2017-2018 (c). There are no estimates for 2018-2019 as the PWRF simulation which is used as**
**input to the COSIPY SMB model was only available until December 2018. The dark black contour marks 1000m a.s.l and the**
**grey contours are every 100m.**
**3.4 Surface mass balance**
To assess whether high areas of SGL development relate to the Surface Mass Balance (SMB), the COSIPY
SMB estimates from Blau et al. (2021) are used. COSIPY has been previously tested for a number of glaciers in Tibet
(Sauter et al. 2020) and evaluated for 79 °N Glacier by Blau et al. (2021). The SMB estimates from September to the
following August for 2015 to 2018 are shown in Figure 9 (2018 to 2019 was not simulated, as COSIPY uses the PWRF
output as atmospheric input). Spatially, the SMB is similar in 2015/2016 to 2016/2017, despite the warmer summer of
2016. Low-lying areas of the 79 °N Glacier tongue, Zachariae Glacier and areas up to 1000 m a.s.l. were in a negative
SMB area in 2015/2016. The following year, the negative SMB extends further inland and to higher altitudes up to
1300m a.s.l (Figure 9). The similarity in SMB between 2015/2016 and 2016/2017 is further presented in Figure 10.
Vertically, the annual SMB profiles are similar in 2015/2016 and 2016/2017 with a negative SMB up to 1400 m a.s.l
(Figure 10a). The summer SMB remains negative up to elevations of 1600 m a.s.l. for both 2016 and 2017, which
coincides with the approximate maximum elevations of SGLs in these years (Figure 4a, b). The annual and summer
SMB in 2018 is considerably different to the previous two years. The annual SMB is negative only at elevations less

than 400 m (Figure 10), which is restricted to areas of the floating tongue only (Figure 9). The summer SMB is also only negative up to 1000 m a.s.l. which also pinpoints the maximum elevation of SGLs in 2018 (Figure 4c).

It is likely that expansion of melt ponds at higher elevations is partly controlled by spikes in the SMB immediately prior to pond development, especially towards the end of the melt season. In summer 2017, SGL development at higher elevations occurred later in the melt season (Figure 3), despite the daily Ta already falling below 0°C. The week prior to July 20th, 2017 (Figure 3a), SMB was mostly positive at elevations greater than 900 m (Figure S4a), however for the five days prior to August 23rd, 2017 (Figure 3b), SMB returned to negative at these higher altitudes (Figure S4b), despite an overall trend towards a positive SMB at lower elevations (Figure S4c). Therefore, not only the local meteorology but also the SMB controls the SGL development, especially at higher elevations.

**4 Discussion**

Summer 2016 saw the largest loss of glacier area over the GIS since 2012, which was the standout, record-breaking melt year since records began (Hanna et al. 2014a). More recently, summer 2019 again broke records for melting and temperatures. However, our understanding of the relationship between air temperature and melting is complicated by the development of SGLs and the interaction of other climatic variables. With a projected increase in air temperatures and inland expansion of SGLs into the year 2100 (Leeson et al. 2015; Igneczi et al. 2016), it is important to understand the linkages between different climatic variables and the spatial distribution and temporal evolution of SGLs in the northeast of Greenland. Furthermore, the role of supra glacial melting within the glacial-hydrologic system is in need of further assessment. In a number of studies, enhanced surface melting has contributed to accelerated glacier velocity (Bartholomew at al. 2011; Rathmann et al. 2017), however in other Greenlandic glaciers, especially land-terminating glaciers, ice velocity has decreased with warmer summers (Sundal et al. 2011; Tedstone et al. 2015).

The spatial spread of the SGLs on 79 °N Glacier from lower to higher elevations as the melt season progresses is also seen on Leverett Glacier in southwest Greenland (Bartholomew et al. 2011). In southwest Greenland, as the melt season develops, runoff from up-glacier (higher elevation) regions contributes to subglacial discharge at the base of the land-terminating glacier, due to a larger melt area and higher air temperatures (Bartholomew et al. 2011). A similar transport of melt water from surface to base is hypothesised for 79 °N Glacier also. Rathmann et al. (2017) observed a seasonal increase in ice velocity following the particularly warm summer of 2016. An extension of the Rathmann et al. (2017) study and estimation of the volume of water potentially interacting with the base of the glacier is an important area of future research.

The rapid increase in SGL area over 79 °N Glacier during June in most years was also observed at Petermann Glacier at 81 °N in the northwest of Greenland. Other similarities in SGL characteristics are found between 79 °N and Petermann Glaciers, including the spatial distribution of the SGLs and the onset of above-freezing air temperatures at the start of June (Macdonald et al. 2018). The only summer with overlap between the current study and the Macdonald et al. (2018) study is 2016. In both locations, this year was characterised by larger total SGL area and warmer than average air temperatures, highlighting the relationship between SGL development and climatic factors across the north of Greenland. However, as noted by Macdonald et al. (2018) and observed in the current study in 2018, the low elevation of these regions dictates that even in cool years, SGLs still form on the lower section of the glaciers.

Langley et al. (2016) hypothesized that SGL expansion in the early part of the season is particularly rapid, as even small changes in air temperature can increase the total lake area. A rapid increase in lake area was seen at the start of the 2016 and 2019 melt season over 79 °N glacier, however in 2017, late-summer temperatures led to later expansion of SGLs. The large rate of increase at the start of summer 2016 (Figure 2) is likely skewed by the slightly lower temporal resolution in 2016 (approximately 3-7 days) compared to the other years (1-2 day). In 2016 and 2017, there

was a lower temporal coverage than the following years as only one Sentinel satellite was in orbit and data quality was
poorer (Hochreuther et al. 2021). However, upon visual inspection of the satellite images, 2016 also saw a rapid
expansion in SGLs, similar to 2019.


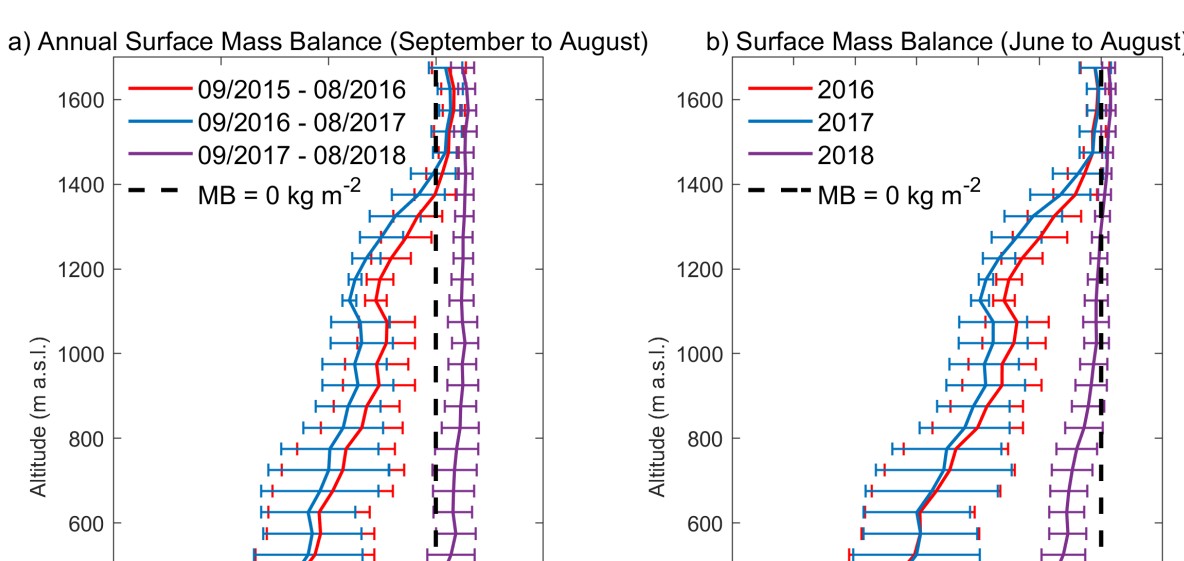


**Figure 10: The annual (a) and summer (JJA) surface mass balance between September 2015 and August 2018, averaged over**
**each altitude in 50m bands. Error bars indicate the standard deviation of SMB for each grid in the respective altitude band.**

Warmer summer air temperatures alone do not always lead to enhanced melting. For the Shackleton ice shelf
in Antarctica, the years with largest SGL area and volume were not always in the same years as the highest summer
near-surface air temperatures (Arthur et al. 2020). With only four years of data in the present study, no major
conclusions can be drawn on this, however it is clear that precipitation also had an impact on SGL area and
development over northeast Greenland. In Buzzard (2018b), the relationship between snowfall and melt pond depth was
not simple or linear. A small amount of snowfall will promote melt pond development, as there is more water available
at the surface, however a high amount of accumulation can bury the melt pond (especially if the surface has started to
freeze towards the end of summer) and reduce melting (Buzzard et al 2018b). We also see evidence of this non-linear
response. A combination of a large amount of snowfall prior to the 2018 melt season, and below average summer air
temperatures led to a lower total area of SGLs and positive mass balance over the majority of the glacierized area
(Figure 9, 10). With a thicker snowpack, it took longer for the SGLs to form, as there was more pore space for water to
percolate through before pooling. A thick snowpack was also responsible for a 20-day delay between above-zero air
temperatures and runoff in the southwest of Greenland, as the meltwater initially refroze within the cold snowpack
(Bartholomew et al. 2011). In the present study, the duration between above-zero air temperatures and melt pond
development varies from 7 to 14 days at KPC_L and 1 to 16 days at KPC_U. The shortest duration was observed in

2019, which had the thinnest snowpack of the four-year period. Similar conclusions were found for Tibeten glaciers (Mölg et al. 2012) and Shackleton ice shelf in the Antarctic (Arthur et al. 2020). ~~Arthur et al. (2020) found that higher accumulation rates contribute to higher firn air content, which allows more water to be retained within the snowpack rather than pooling into SGLs. The higher snowfall in 2018 also created the conditions which allowed the lakes to remain open for longer than in previous years, because more water was available towards the end of the melt season.~~

Conversely, the year with the smallest snowfall amount (2018-2019 accumulation season) was not followed by the summer with the fewest melt ponds. However, the much higher air temperatures and late summer freeze up of SGLs in 2018 played a bigger role. Summer 2016 saw the second largest total SGL area and spatial distribution of SGLs. This year also saw a large amount of precipitation fall as rainfall in summer. Rainfall is additional liquid for the surface of the glacier, provides heat to the snowpack and refreezes into solid ice lenses which preconditions the glacier surface for further SGL development (Machguth et al. 2016). Rainfall associated with summer storms has been linked to extreme melting events in southern Greenland by Oltmanns et al. (2019) and enhanced ice velocity in western Greenland by Doyle et al. (2015). Similarly, Tedesco and Fettweis (2020) concluded that low snow accumulation was also partly responsible for the extensive melting along much of the coast of Greenland in 2019.

Relationships between large-scale temporal and spatial anomalies within the atmosphere, termed teleconnections, have been found to influence the climate and mass balance of Greenland (Tedesco et al. 2013; Lim et al. 2016). With only four years of data in the current study, we are unable to draw conclusions about the role of teleconnections in the development of SGLs, however there is evidence that combinations of teleconnection indices play a role in the near-surface climate and therefore SGL development in the northeast of Greenland. In 2016 and 2019, the average summer (JJA) North Atlantic Oscillation (NAO) index was strongly negative (-1.36 for 2016, -1.23 for 2019) (see Supplementary material for teleconnection data). Simultaneously, both the summer East Atlantic (EA) index and the Greenland Blocking Index (GBI) were strongly positive in both of these years. In summer 2016, the EA (GBI) summer average was 1.44 (1.73). Similarly, in 2019 the JJA average EA index (GBI index) was 1.1 (2.26). This combination of strong negative NAO and strong positive EA also occurred in both summer 2010 and 2012, when extensive melting was observed over the GIS (Lim et al. 2016). In terms of teleconnections, the biggest differences between 2016/2019 and the 2017/2018 summers was the NAO and GBI summer indices. In 2017 the NAO index was positive in June and July. In 2018 the summer NAO index was strongly positive (1.74), with all summer months observing a positive NAO signal. The GBI for summer 2017 and 2018 was weakly negative (-0.03) and negative (-0.57) respectively. In terms of the teleconnection indices evaluated here, summer 2017 appears to the be the intermediate or transition year between a particularly strong negative NAO in 2016 and a strong positive NAO in 2018. A decreasing trend in summer NAO since 1981 has been previously identified and is believed to be partly responsible for record-breaking warm temperatures over Greenland in the most recent decade (Hanna et al. 2014).

The relationship between teleconnections and precipitation is more complicated and is often only significant in the southern part of Greenland where the majority of the precipitation falls. Bjork et al. (2018) identified a positive relationship between NAO and precipitation in eastern Greenland: there is more precipitation during positive NAO years. The year with the largest cumulative precipitation amounts was the 2017-2018 accumulation season, which was also characterised by a strong positive NAO index. However, the relationship between NAO and precipitation for NE Greenland cannot be assessed with certainty in this study.

Although we present only four years of results here and previous studies in this region are sparse, we are confident that SGLs are a persistent feature in the NEGIS and 79 °N region. Sundal et al. (2009) observed SGLs between 2003 and 2007 using MODIS data. With the availability of very-high resolution (10 m) Sentinel data, the SGL areas are less erroneous than previously stated using lower-resolution MODIS data (250 m) (Hochreuther et al. 2021).

There is an increase in the maximum altitude of SGL detection between the early 2000's study of Sundal et al. (2009) (1200 m a.s.l) and the results presented here (1600 m a.s.l). The lakes at these higher elevations are larger and therefore would have been detected by the MODIS data in the Sundal et al. (2009) study, were they present. Therefore, it is likely that maximum lake altitude has increased over time.

This is not surprising given an increasing air temperature trend of 0.8 °C decade$^{-1}$ over 79 °N Glacier (Turton et al. 2019), and model suggestions of inland expansion in this area into the 21$^{st}$ century (Ignéczi et al. 2016). Leeson et al. (2015) concluded that maximum lake altitude could reach up to 2221 m a.s.l with RCP 8.5 future projections. Although there are a number of assumptions made in our comparison to Sundal et al. (2009), it is possible that inland expansion of lakes is occurring under increased air temperatures in this region.

Under certain high-melt years, surface rivers have been observed for a number of northern Greenland glaciers, including 79 °N (Bell et al. 2017). While we do not consider meltwater channels in our analysis and focus only on SGLs, a number of linear features similar to rivers are clearly visible in the Sentinel data (Figure S1). This highlights that more liquid water is likely present on and within the glacier than discussed here. There is even some evidence of the persistence of liquid water in melt lakes during the winter season on 79 °N Glacier (Schröder et al. 2020). ~~used Sentinel 1 SAR data which can detect water without the presence of sunlight (unlike optical sensors such as Sentinel 2) and under the snow surface.~~ It is hypothesised that lakes beneath the surface were formed in particularly warm years (such as 2019) and then subsequently covered by a thin ice lens or snow (Schröder et al. 2020).

Estimates of the SGL volume are not provided in this study, which is unusual for these types of studies (e.g Pope et al. 2016., Arthur et al. 2020). We hypothesise that SGLs in this region are much deeper than those observed in the west of Greenland (on the order of 0-10 m). Neckel et al. (2020) recorded the depth of an SGL on the 79 °N Glacier, which, at the edge of the lake, had a depth of 10.8 m. The same lake drained suddenly in September 2017, and analysis of the height difference from a full to empty lake using DEMs revealed a subsidence of 50 m in the centre of the lake (Neckel et al. 2020). Therefore, applying the same albedo-depth calculation to the lakes in northeast Greenland as in western Greenland would largely underestimate the volumes. In-situ observations of these lakes are required to calculate depth and volume with a different albedo-depth coefficient. Fieldwork is planned for this region, to observe the depth of SGLs.

**5 Conclusions**

In this study we provide a multi-year analysis of the area of SGLs over the 79 °N Glacier (northeast Greenland) and investigate the atmospheric and topographic controls of the evolution of the lakes. SGLs have been automatically detected using Sentinel-2 data, from 2016 to 2019. The melt detection algorithm implemented here and developed by Hochreuther et al. (2021) is automated, meaning that this work can be continued in the future to analyse a long-term time series of SGL evolution. Our findings would ideally now be expanded to include volume estimates and to model the surface and subglacial hydrology to provide an estimate of the volume of fresh water entering the ocean.

Whilst the SGL location is primarily determined by topographic depressions and the slope of the ice sheet, the occurrence of lakes within these depressions relies on the local meteorology and SMB. Similar to the spatial distribution, the maximum size of individual lakes is controlled by topography. At higher elevations, larger lakes form due to a lower slope angle (Figure 4). The larger total SGL areas in 2016 and 2019 were due to lakes developing at higher elevations than in 2017 and 2018, as opposed to individual lakes becoming larger. SGLs refreeze and melt in the same locations above the grounding line each year, but maximum inland expansion of the lakes depends on climatic conditions. ~~Schröder et al. (2020) state that liquid water remains in the lakes throughout the year but can become buried~~

~~by an ice lens or snow which prohibits the detection by optical sensors. It is Therefore, in warmer years such as 2019,~~
~~the snowpack is melted to reveal melt lakes formed previously, which contributes to the larger lake area.~~

The two key climatic variables controlling the development of the SGLs are air temperature and snowfall.
Below average air temperatures and high snowfall accumulation prior to the melt season of 2018 contributed to reduced
lake extent, a reduced amplitude in the seasonal cycle of lake evolution and late season freeze up of the SGLs. These
climatic conditions led to a largely positive mass balance at all altitudes except the very lowest lying regions.
Conversely, in the prior two years, surface mass balance was negative for a large portion of 79 °N Glacier and the
surrounding area. Largely this was driven by the above average air temperature, evident in both the in-situ AWS data
(Figure 5) and in the regional atmospheric modelling output (Figure 6, 7). The duration between onset of above-zero air
temperatures and SGL development varies between 1 and 16 days, depending on the year and elevation. The snowpack
thickness prior to the warm air temperatures likely also has an influence on this duration.

The role of clouds in melt production over the Greenland Ice Sheet is often studied (e.g Bennartz et al. 2013).
Within the four years, the warm summer of 2016 coincided with a positive bias in SWin and a reduction in cloud cover,
however the warm June of 2019 was characterised by negative biases in SWin. Similarly, the relatively cool summer of
2018 was characterised by positive anomalies in SWin and higher than average cloud cover in June. With just four
years of data in the current study, no clear conclusions can be drawn on the role of clouds on the development of SGLs
in this region.

Whilst 2019 was record breaking in terms of melt over much of the Greenland ice sheet, in fact second only to
2012 (Tedesco and Fettweis, 2020), the summer of 2016 was only warm and extreme in the northeast region. The
extreme summer temperatures led to extensive SGL formation over the 79 °N Glacier, as well as subsequent ice
velocity acceleration (Rathmann et al. 2017). Similarly, 2019 was not a record-breaking melt year in the northeast of
Greenland, and at lower elevations, the number of melt days (TSK$_{melt}$) and the duration of melting was less than in other
years. This highlights the importance of regional studies of extreme melting, as well as the Greenland ice sheet-wide
studies.

There is some evidence of inland expansion of lakes between the Sundal et al. (2009) study, which looked at
SGLs between 2003 and 2007, and the present findings from 2016 to 2019. The highest elevation of SGLs in the early
2000's was 1200 m a.s.l, whereas in the late 2010's, SGLs above 1600 m a.s.l were observed. This is in line with global
climate model projections for inland expansion of SGLs and the ablation zone under climate change (Ignéczi et al.
2016). The northeast of Greenland is expected to undergo the largest changes in SMB and SGLs by 2100 and therefore
should feature in future atmosphere-glaciohydrology studies.

**6 Data Availability**

The daily average surface mass balance data is available at: https://doi.org/10.5281/zenodo.4434259. For higher
temporal resolution see Blau et al. (2021). The daily average PWRF data is available at:
https://doi.org/10.17605/OSF.IO/53E6Z. For higher temporal resolutions see Turton et al. (2020). Lake outline
polygons and cloud masks are available on request and are currently being uploaded to Pangaea Data Centre, pending a
DOI.

**7 Author Contribution**

J.V.T wrote the manuscript and conducted the climatological analysis. P.H developed and applied the automatic
detection algorithm for the SGLs and assisted in discussing the results. N.R assisted in the development of the algorithm
and writing the manuscript. M.T.B. conducted the SMB modelling and analysis.

## 8 Competing Interests

The authors declare no conflict of interest.

## 9 Acknowledgements

We are grateful to the European Space Agency (ESA) for providing the Sentinel-2 data and to the Greenland and Denmark Geological Survey (GEUS) for maintaining the AWS and providing the data. We acknowledge the German Federal Ministry for Education and Research (BMBF) for funding this work as part of the GROCE project (Greenland Ice Sheet/Ocean Interaction) (Grant 03F0778F and 03F0855F). We also thank the High-Performance Computing Centre (HPC) at the University of Erlangen-Nürnberg's Regional Computation Centre (RRZE) for their support and resources. We also thank two anonymous reviewers and the editor Dr Stef Lhermitte for their insights and feedback.

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
