# Peer review of "The distribution and evolution of supraglacial lakes on the 79 °N"

_The Cryosphere, 2021_

## Referee Comment (RC2)

Using Sentinel-2 imagery and a water identification algorithm, the authors analyze the seasonal evolution for the years 2014 to 2019 at the 79N glacier in northeast Greenland. They analyze their results alongside topographic, climatic and surface mass balance data to assess the role of these factors in lake evolution.

While there have been numerous studies of lake evolution in Greenland, this study focuses on an area of the ice sheet where there are few studies of supraglacial lakes. The study also devotes substantial attention to the role of climatic factors in lake development, in a way that many previously published papers do not. Therefore, I believe that this paper will be of substantial interest to the scientific community.

**Key Comments**

There are some key areas/issues that I think could be improved in a revision:

- I would like to see more explanation and justification of the lake identification method. A brief explanation and reference is provided within, but I think more detail needs to be provided, especially with it being a new approach. Key things I'd like to see are
    - Why is this method used rather than other often-used NDWI or band-thresholding approaches? Reference to Williamson et al. (2017), which assesses various methods, would be appropriate here.
    - How exactly are the bands used to distinguish water from ice/slush?
    - What are key limitations? Will some slush be falsely identified as water? Will some streams be identified as lakes or is there a shape criteria to avoid this?
    - Why is depth not calculated?
    - Why are lakes below 0.015km2 not included?

- Following from that, given that the average size of lakes in 2018 is 0.02km2, eliminating lakes below 0.015km2 seems like it will lead to a substantial amount of lakes being missed, distorting some analysis. I suggest considering a lower cut-off. If not, this limitation needs to be stated.

- The discussion leads off with, and primarily focuses on, comparison with Antarctic studies. Comparison with Antarctic studies is worthwhile too, but I would be more interested in comparison with other areas in Greenland. The well-studied W/SW (e.g. Miles et al 2017, and numerous others), and Petermann Glacier in northwest Greenland, at a similar latitude to this study (Macdonald et al., 2018).

- One of the key interests regarding supra glacial lakes is how they drain, especially given the role this can play in dynamics, but there is little mention of it in this study. It is okay that if drainage speed or mechanism is not systematically studied, but it would be good to make some comment on your assessment of how lakes drain based on your observations.

- I think a few figures need substantial work to make the study clearer - Fig. 3 and 1 in particular. More below.

- There are parts where the structure and paragraphs needs some review. I ask the authors to review this generally - assess whether material is in the correct section (i.e. intro v results v discussion v conclusion), and check for sentences that seem out of place with theme of paragraph. Some key suggested changes:
    - The NAO is not mentioned at all in the introduction and then takes substantial space in the discussion. I think this needs to be indicated as a theme in the intro given its

significance - some of its discussion in the discussion section can be put in the intro.
- A large part of the conclusion seems out of place. New discussion is brought in - e.g. the Neckel et al. reference part

- I think the conclusion needs substantial work. In addition to the above comment, in addition to the summaries of each year, I think the key take aways need to be highlighted and, importantly, there should be some discussion of the wider implications of the findings.

A small suggestion: perhaps it would be worth looking at the role of atmospheric rivers. See Mattingly et al. (2020) - all references at the bottom.

Overall I very much enjoyed reading this valuable study and hope that these points can help the authors to make the most out of the hard work they have put in. I provide line-by-line remarks/suggestions below.

**Line-by-line and figure comments**

L34 - "meltwater drainage channels" ?

L36 - "and therefore **they** absorb…"

L47 - "have investigated **the relationship between** the seasonal evolution…"

L55 - Sundal et al. say "Since the ASTER image used for the test was acquired at a later stage in the melt season, the percentage of unidentified **lake area at the start of the summer is likely to be somewhat higher than 12%**"

L56-58 - this sentence about Sentinel 1 SAR seems out of place and without explanation, since the next sentence is instead about Sentinel 2.

L60-61 - this sentence about what you do in this study does not fit well with the structure of the intro. It led me to think you were about to elaborate on your own approach, but you go back to background discussion.

L66 - please provide some elaboration on the point about teleconnection signals.

L79 - "whether **and how**"

L93 - it would be more useful to give the approximate return time at high latitudes than the equator

L167-168 - Specify for where this fact refers to. Also, I suggest leading the paragraph/section with your results and mentioning this later - it seems odd to lead a results section with results from the literature.

~L175-180 - I note an inconsistent use of tense. I don't mind if in present or past tense, but check for consistency.

L178 - Careful to be specific. Does July refer to July in all years? Have just been referring to June 2017 so not totally clear. Check always clear with this. Also consider starting a new paragraph here.

L203 - "largest peak **total** SGL area" - check clear on this sort of thing in all instances

L204 - "smallest peak **total** SGL area"

L207-209 - Rephrase for clarity. I think part of issue is 'close up' can be read as in 'close up to something'.

L215-216 - Rephrase - something like "Given sufficient meltwater availability, the location of lake formation…"

L223-4 - You say earlier that you don't identify lakes below the grounding line due to issues with lack of DEM and them moving. So I assume this is based on visual analysis? If so, mention in the methods that you will do this.

L223 - "the lakes move position laterally" is not clear. Do you mean they advect downglacier with the flow of the glacier (e.g. as in Macdonald et al 2018 and Langley et al 2016)?

L225-226 - please rephrase this for clarity

L233-235 - it seems odd to make a first mention of depth here. If you make some assessment of depth in this way, you should mention it in the methods section. Also, I'm uncertain what 'not shown' means - I think not in any of the figures? It would be good to at least show an example in a supplemental figure. If you need to convey 'not shown' anywhere, please state what you mean more specifically.

L236-238 - this paragraph seems out of place.

L246 - I think for this audience you should explain what 'skin temperature' means

L270 - "agreement **with**"

L430-L432 - Please be explicit about what your point is here. I assume you're suggesting that the calving event possibly implies warm temperatures?

L469 - Do you mean to say "Not only local meteorology …but also SMB prior to pond development". A reader might think from the sentence structure that you're implying "not only local but regional". Please state.

L472-3 - I believe here you mean across the Greenland Ice Sheet, but since you don't state it might read as if you mean for the study area.

L474 - I would lead with comparison to Greenland, see main points at the start.

L493-4 - "however **a high amount of** accumulation can …and **reduce** melting"

L505 - "second largest **total** (?) SGL area" And I think I know what you mean by 'spread' (a broader area?), but please explain

L555 - ~"While we do not consider meltwater channels in our analysis and focus only on SGLs…"

L558-61 - state where this Schröder study focuses on. And I do not think you need to explain here that S1 SAR works without sunlight.

L564 - Given the title of the paper refers to 79N Glacier, it seems odd to summarize the study as being of the 'North East Greenland Ice Stream'

L573-75 - leave out the reference to Schröder in conclusion, leave that to earlier sections, and just make your assessment here.

L580-81 - This sentence about volume measurements is out of place in the paragraph and section

**Fig1**
Please provide a small map showing location within Greenland. (Newly-released QGreenland may help with this).
Please mark on the groundling line.
Please label the 'NE Greenland Ice Stream'. This is mentioned in paper but it's not clear how it fits in with 79N Glacier.
I suggest showing and labelling the wider study area in a satellite image/mosaic. Then show terrain height on that, or make it a separate panel.
Is it important to show calving in main figure? The calving event is a very minor part of the paper.

**Fig2**
Since you refer to months in the text, please put them on the x axis.
I find this color scale difficult - the change from 0 to 300 isn't a lot, making it hard to decipher medium or smaller changes.

**Fig3**
I find it very difficult to make an assessment of what is happening in this figure without spending a while looking at it. Perhaps consider having a zoomed in sample site. If you use this approach please label July/August on the figures themselves, and consider using a satellite images as the background (though I understand that may not work if other surface features make it busy)

**Fig5**
Can you make the scale consistent with Fig2? If not the scale itself, please make the labels consistent (i.e. add day of year, and add month labels to Fig2)

**References**

Bartholomew et al (2011) - https://agupubs.onlinelibrary.wiley.com/doi/full/10.1029/2011GL047063 (not mentioned above but seems particularly relevant, as looks at observations of temperature and upglacier spread of lake drainages)

Macdonald et al (2018) - https://www.cambridge.org/core/journals/annals-of-glaciology/article/seasonal-evolution-of-supraglacial-lakes-on-a-floating-ice-tongue-petermann-glacier-greenland/3FB0176ABB47735A10DFB8F72E47534D

Mattingly et al (2020) -  https://journals.ametsoc.org/view/journals/clim/33/16/jcliD190835.xml

Miles et al (2017) https://www.frontiersin.org/articles/10.3389/feart.2017.00058/full?sa=X&ved=2ahUKEwiL-eX06pHnAhUPnq0KHarTC7oQuAIwLXoECBIQAg

Williamson et al (2017) https://www.sciencedirect.com/science/article/pii/S0034425717301918

---

## Author Response (AR1)

Response to reviewer 1:

Dear reviewer, thank you for your feedback and suggestions to improve our manuscript. We have taken the vast majority of your suggestions on board and together with reviewer 2s feedback, we believe our manuscript is now much stronger and appreciate your efforts. Below we provide explanation and changes to the larger points that you suggested, as well as point-by-point responses to the smaller changes. We have written our response in purple and where direct quotes from the manuscript are written, we use italics. A tracked-changes manuscript has also been uploaded, with new sections in red font and lines through removed sections. In addition, we have provided line numbers to our feedback below where possible.

**Specific Comments**

I struggled to follow Section 3.3 - partly because it can be quite dense but mostly, I believe, because it strays from the narrative structure of the rest of the results. Sections 3.1, 3.2, and 3.4 each address individual variables (lakes, topography, and SMB respectively), describing how each of these properties vary between 2016-2019. In contrast, section 3.3.1-3.3.4 each address individual years (2016 to 2019 respectively), describing how a variety of properties (temperature, SWin, precipitation) behave in each year. This abrupt inversion to the logical structure of narrative makes it difficult to follow how the climatic variables change across years. The structure should be consistent across the results, and I believe the paper would be better served by continuing to treat variables separately, highlighting the narrative of interannual change. This would further benefit the paper by making sure each year and variable receives equal treatment: for instance, section 3.3.1 (2016) deals with observational $T_a$, $T_{SK}$, and onset, as well as PWRF data, but section 3.3.4 (2019) discusses only observational $T_a$ (for what it's worth, I found section 3.3.4 more focussed and easier to follow).

Thank you for your comments and suggestions. We have now edited the structure of section 3.3 to focus on individual climatic variables as you suggest. It is not however possible to focus equally on the variables, as neither SMB nor PWRF data is available for 2019, which perhaps led to this section being easier to follow. However, we have now tried to focus the manuscript in this section and in response, we believe it is now easier to read. Please read the updated section and feel free to suggest more modifications if you still struggle with this section.

A key point of interannual comparison is the total area of lakes, but the way this is visualised in Figure 2 is hard to interpret. It would be easier to follow a line graph, which would be simple to add as a second axis of the Figure 2 panels. Additionally, it might be useful to present lake area data alongside the later data, so that the reader does not have to continually refer back to earlier figures/text on lake development. One way (although I don't mean to prescribe here, so certainly not the only or best way) of doing this might be to split Figures 5 and 8 into four panels - each representing individual years (mirroring Figure 2) - and including lake area as a second axis. Splitting Figure 5 into panels of individual years would also make it possible to include, perhaps as vertical lines or shaded boxes, the data presented in Table 2 (periods of $T_a > 0 \overset{..}{E}$ C and melt ponding). This work would allow for the easy visualisation and comparison of a lot

of data that is currently only accessible via text or table, or by comparing disparate figures and tables on different pages. In turn, the authors may be able to simplify much of the denser text that is currently spent describing the temporal variations in these data.

Thank you for your comments. We have tried to incorporate many of your suggestions or altered the figures to allow better comparison between years and locations, and to remove the need for readers to move backwards and forwards in the manuscript. Please also see the changes suggested by reviewer 2.

Specifically, Figure 2 now includes a line graph with total lake area, as well as lake change rate (which no longer uses a colourbar). Figure 3 is split into 4 panels to show annual variability. Figure 5 has now been split by year into four panels also, containing both KPC_L and KPC_U information and including SGL opening and closing dates from Table 2. We have kept table 2 in the manuscript so that the exact dates are still available. As this figure is now quite busy, we have not added lake area, but this information is now on Figure 2. We have not split Figure 8 by year, as you suggested, as there is only limited information on this figure, and it covers more than one calendar year (accumulation period is spread from September to May).

The results tend to treat lake behaviour in bulk, rather than considering the behaviour and/or heterogeneity of individual lakes. I think this is largely fine for the purposes of the study, but it would be nice to see some consideration at specific points.

Thank you for your suggestion. As there are so many lakes present, it is difficult to discuss individual lakes. However, we have included discussion of specific lakes in terms of lake drainage now, with additional figures in the supplement.

First, Section 3.1 discusses late-season increase in lake area. Is this a general trend common to all lakes (due to, e.g., a melt day / rainfall event)? Or is it a result of a few individual lakes rapidly increasing in area (due to, e.g., reorganisation of the surface drainage system). If the latter, does this also occur at other points, but is masked in the bulk data?

The late-season increases are only seen in 2017 and 2018 and are a response to climatic factors. In 2017, there was a period of warmer than average conditions at the end of August which led to a few lakes re-opening at mid-level elevations. In 2018, the total lake area is quite small, so any small changes in lake area have a larger impact. This late-season increase is a small, short-lived peak which is due to the warmer conditions and rainfall events in August. We have altered this sentence to make it clear that this isn't a trend which occurs in every year. '*However, in some years there can still be individual days of increasing SGL area (positive change rate) punctuating the overall decline in SGL area towards the end of the melt season (Figure 2). This can occur due to periods of warm air temperature or late-season rainfall events.*'

In 2019 there were periods of rapid decline in lake area, which are attributed to drainage events. We have now included a section which presents these findings also (Line 271 onwards): '*Significant decreases of total lake area can be attributed either to sudden climatic changes, or to consecutive drainage events. In 2019, the sudden decrease around DOY 240 is attributed to a large freeze-over of the majority of all lakes above 700 m a.s.l. Conversely, the*

*decrease following the 2019 peak of total lake area on August 2nd (DOY 214) was caused by a step-wise drainage pattern, starting with larger lakes at high altitudes, followed by drainage events close to the ice front of Zachariae and accompanied by a speedup of calving and seawater movement (Figure S3).'*

Second, it is not discussed how the lakes are draining (rapid vs slow drainage). Can the authors quantify (or at the very least, comment qualitatively on) the relative dominance of drainage modes? Recent work has begun to take an interest in how lake drainage may differ between land- and marine-terminating sectors of the Greenland Ice Sheet (e.g. Williamson et al. 2018a, Chudley et al. 2019), but this is still largely focussed in the SW/W of Greenland. Further observations from this unique sector would be of considerable interest.

We understand the interest in lake drainage, as it is closely linked to ice velocities and thus to calving etc. Though the focus of the paper is centred around the relationship cryosphere – atmosphere, we added information about drainage patterns and over-freezing influencing the total lake area, as well as a new figure in the supplement with an example of a series of rapid drainage events within 96 hours. Regarding individual rapid drainage events, we agree that data from this sector of Greenland is of high interest and take this as an inspiration to analyse this particular matter more in depth in an additional paper. For this study, we think we have to limit the examination to the bulk level data in order to keep this part in line with the other sections and the scope.

On a different note, considering how much time is spent considering the influence of teleconnections in the discussion, I am surprised by their relative lack of attention in the abstract, introduction, and conclusion (and perhaps, also, their lack of inclusion graphically). On a basic level of critique, this means that their inclusion in the discussion comes a bit out of nowhere. However, more interestingly, I do not think that many (any?) observational studies of lake variation consider these modes of climate variability, so some time spent introducing them and their context may be useful for those coming from other areas of the discipline (e.g. optical remote sensing / computer vision), as well as highlighting them in the abstract/conclusion so that the interesting conclusions of this paper can reach the widest audience!

With just four years of data for comparison, it is difficult to draw strong conclusions to teleconnections, but we wanted to include them in the discussion due to the likely importance of such indices for record melting events over Greenland. We have now included a section in the introduction (most of the background information about teleconnections from the discussion has been moved to the introduction). We agree that introducing them earlier on is better for a wider audience understanding and interest, so thank you for pointing this out.

**Minor Comments**

Abstract

- L10: "Together with two neighbouring glaciers…". Perhaps a broad statistic for a study focussing on just 79N? Why not just say 8%, as per L25?

  Changed to 8% and removed the 'together with two neighbouring glaciers'.

- L16: "2014 to 2019". The primary ablation seasons considered are 2016-2019, so this might set expectations high. Somehow make it clear that whilst you examine lakes over 2016-2019 melt seasons, you examine influences up to however many months prior

  This is actually a typo and should say 2016-2019, so we have now changed this. Thank you.

- L18: "Over 1400 m" > "Up to [x] m"

  Changed to 'up to 1600 m'.

Intro

- L34 "Low-elevation" is subjective, considering lakes can extend away from the outlet glaciers and up onto the ice sheet proper. Perhaps just 'on the ablation zone of the ice sheet'?

  We have removed 'low-elevation' and just kept glaciers, as the melt ponds also extend into the accumulation zone.

- L38-41 This paragraph focuses on short-term accelerations: perhaps make it clearer that meltwater has also been shown to have negative feedbacks on ice velocity in the medium-long (seasonal-decadal) timescales, at least in land-terminating sectors (e.g. Tedstone et al. 2015; Sundal et al. 2011, etc.).

  Thank you for your suggestion. This section now reads as: *'Ice velocity increases and decreases have been linked to drainage of SGLs across Greenland. Short-lived velocity increases have been observed during summer in several marine-terminating glaciers, including 79 °N (Rathmann et al 2017). Both Rathmann et al (2017) and Vijay et al (2019) hypothesise that the summer speed-up of 79 °N occurs when SGLs drain to the base and alter the subglacial hydrology. Conversely, on land-terminating glaciers, SGL drainage has been shown to reduce ice velocity in the seasonal to decadal time scales (Sundal et al. 2011; Tedstone et al. 2015).'*

- L55-56 "...likely underestimated the lake area by 12%". Was this shown in the Sundal et al. 2009, or did a later high-resolution study quantify this? A bit unclear as written.

  This was shown by Sundal. We have re-written it to: '*Sundal et al. (2009) used MODIS data to assess the lake area between 2003 and 2007 for 79°N amongst other locations. However, as the ASTER images were acquired at a later stage in the melt season, the percentage of unidentified lake area at the start of the summer is likely to be higher than 12% (Sundal et al. 2009).'*

- L58-59 Willamson et al. 2018b also provide a prior example of Sentinel-2 automated lake detection.

Here, we are talking specifically about the northeast Greenland region. We have however included this study in the first paragraph of the data and methods section now.

- L80 I always understood VHR to be ~1m resolution - does Sentinel-2 really make the cut?

  As I am primarily an atmospheric modeller, 10m resolution is very high resolution! But we do use 'very' to separate the higher resolution of sentinel (10m) to the PWRF model (1km)

Data and Methods

- L91-95 This introduction to the Sentinel-2 mission is generously detailed, and can probably be safely removed if the intention is to provide a brief overview (as per L91).

  We have removed some aspects, but also elaborated in other sections to fit the requests of both reviewers, please see the revised manuscript.

- L100-101 It would be nice to at least reference by name the pre-processing steps applied, so that those familiar with the techniques do not have to consult another paper. Those who are still interested in the nuts and bolts then have the option to delve further.

  This is tricky, as several aspects of the methodology do not have a name per se, at least not beyond what is already stated. As reviewer 2 requested further details about the methods in the paragraph, we hope that the updated information is now more complete even for readers with a strong background in multi-spectral remote sensing of the cryosphere.

- Paragraph beginning L108 - From the paragraph it seems that the data was cropped to the grounded ice, but how was the spatial limit determined? A fixed outline (e.g. GIMP), user-determined annual grounding lines, etc.?

  Thank you for spotting this! Yes, indeed the data was cropped to the grounded ice before applying various post-processing steps. The GIMP mask was used for this, as you assumed. We added this information to the paragraph, together with the source of the grounding line estimation.

Results

- L170: These lakes are all rather small, quite close to the cutoff size of 0.015 km^2. Could the dataset be sensitive to this chosen limit?

  Potentially it is, regarding the total number of lakes being detected. Regarding the total area, small lakes do not contribute significantly to this number, it is mainly influenced by the largest quantile of lakes. Thus, also general tendencies (e.g. the large difference between years, the timing and duration of lake development etc.) are largely independent of these small lakes. The error is there, but it should be very small compared to the total lake area.

- L174-179: It would be nice to also see absolute as well as relative changes in this text (see also my comments about Figure 2).

  We have now included absolute lake area on Figure 2, and there is a paragraph outlining the absolute lake areas from line 357.

- L80-81: To what extent are these late-season increases spread across all lakes (e.g. due to a high melt day) or more heterogenous (due to, say, reorganisation of the surface drainage system allowing a few lakes to fill)? Is it possible to separate the data by individual lakes?

  Please see the answer to your major comment above in regards to late-season increases.

- L396 - Contextual sentence, probably better belongs in intro or discussion (alongside a citation)

We removed it from the manuscript entirely.

Discussion

- Paragraph beginning L488: Strange absence of reference to any Greenlandic literature here. One of the unique aspects of this study, as identified in the introduction, is the lack of prior SGL studies in the NE of the ice sheet. It is a shame not to see more comparison and contrast to the established SGL literature in the SW and W of the ice sheet.

Thank you for this suggestion. We have now added Greenlandic studies in here and at other points in the discussion. Please see the changed manuscript for specifics.

Conclusion

- Paragraph beginning L577: This paragraph probably belongs at the end of the discussion?

  We have now moved the majority of this section to the discussion. Thank you for the suggestion.

Figures and Tables

- Figure 1: Currently only the land extent is demarcated in Figure 1a - could the ice extent and grounding line(s) be done also?

  Thank you for your suggestion. Along with the comments of reviewer 2, this figure has changed. We now include a map of Greenland, a label for NEGIS and the mosaic of Sentinel granules. The ice extent should now be clear, as is the grounding line.

- Figure 6/7: (i) What do the vectors represent? (ii) Could these two figures be combined into one figure of six panels, so that it is easy for the reader to

compare June/July of the same years as well as individual months of different years?

We considered your suggestion, however when clustered together, the figure panels were too small and the wind vectors were not visible. We think that figures 6/7 should highlight the large interannual difference in the months, rather than the June to July change in individual years. Therefore, we have kept the figures the same as previously, but have included wind vector descriptions in the captions.

- Figure 10 - Notable compression artefacts in this image - possibly just my download, but it doesn't seem to appear elsewhere so thought I'd mention in case it's common.

We have now re-plotted these figures and hope that the compression artefacts are no longer present. Please let us know if you are still seeing this issue in the new version, however. Thank you.

References

Thank you for the suggested references, we have now expanded our manuscript to including discussion of Greenlandic studies. Please see the updated manuscript (especially the discussion) for new citations and references (red text).

Response to reviewer 2

Dear reviewer, thank you for your feedback and suggestions to improve our manuscript. We have taken the vast majority of your suggestions on board and together with reviewer 1s feedback, we believe our manuscript is now much stronger and appreciate your efforts. Below we provide explanation and changes to the larger points that you suggested, as well as point-by-point responses to the smaller changes. We have written our response in purple and where direct quotes from the manuscript are written, we use italics. A tracked-changes manuscript has also been uploaded, with new sections in red font and lines through removed sections. In addition, we have provided line numbers to our feedback below where possible.

Key Comments:
There are some key areas/issues that I think could be improved in a revision:

1) I would like to see more explanation and justification of the lake identification method. A brief explanation and reference is provided within, but I think more detail needs to be provided, especially with it being a new approach. Key things I'd like to see are:

- Why is this method used rather than other often-used NDWI or band-thresholding approaches? Reference to Williamson et al (2017), which assesses various methods, would be appropriate here.
- How exactly are the bands used to distinguish water from ice/slush?

- What are key limitations? Will some slush be falsely identified as water? Will some streams be identified as lakes or is there a shape criteria to avoid this?
- Why is depth not calculated?
- Why are lakes below 0.015km2 not included?

The method that we use has already been published in Hochreuther et al. (2021). In the method paper, the explanation why the static band ratio method was applied was discussed in length, as it is really a key point of the method. Of course, the paper by Williamson et al. (2017) is very relevant, if not central to this issue, and has been referenced therein. Nonetheless, we added further details and the reference to the paragraph to give the interested reader a clearer view on the method without having to consult the reference Hochreuther et al. (2021). Specifically, see lines 170 to 201 for additional information.

2) Following from that, given that the average size of lakes in 2018 is 0.02km2, eliminating lakes below 0.015km2 seems like it will lead to a substantial amount of lakes being missed, distorting some analysis. I suggest considering a lower cut off. If not, this limitation needs to be stated.

The method that we use has already been published, therefore we will not be changing the lower cut-off of the threshold. Though potentially, Sentinel 2 allows a minimum lake size of $100m^2$ (1 pixel), it is not realistic to find a threshold or band combination that separates water from slush/shadow this well, especially in rugged terrain such as crevasse fields, for larger areas. As this is a general limitation of all SGL detection studies and methods, the size of the filter that reduces or eliminates misclassifications ("noise") is a choice based on a fraction of the data. Judging from the sample we used, the threshold of 0.015 km2 is the best possible compromise between false removal of actually present smaller water areas and retaining falsely classified slush/shadows. Lakes this small, even if they are numerous, contribute very little to the total lake area, thus we expect the error that is introduced by the minimum size as being small as well. Apart from the total lake area, the total number of lakes detected is potentially influenced by this threshold, thus we added this information and potential consequences to the paragraph.

3) The discussion leads off with and primarily focuses on, comparison with Antarctic studies. Comparison with Antarctic studies is worthwhile too, but I would be more interested in comparison with other areas in Greenland. The well-studied W/SW (e.g Miles et al, 2017 and numerous others) and Petermann Glacier in northwest Greenland, at a similar latitude to this study (Macdonald et al. 2018).

Thank you for this suggestion. It was indeed an oversight on our part not to include the Greenlandic studies in the discussion. We have now included a number of studies including: Macdonald et al. (2018), Sundal et al. (2011), Bartholomew et al. (2011) and Tedstone et al. (2015). Therefore, Petermann and Leverettt Glaciers are now explicitly named and other areas of Greenland are discussed.

4) One of the key interests regarding supraglacial lakes is how they drain, especially given the role this can play in dynamics, but there is little mention of it in this study. It is okay if drainage speed or mechanism is not systematically studies, but it would be good to make some comment on your assessment of how lakes drain based on your observations.

Thank you for the suggestion. Though lake drainage is not a key focus of this paper, as it above all links lake patterns to ice velocity, we added a paragraph dealing with the influence of rapid drainage events on the total lake area. Additionally, we added a new figure to the supplement, showing a series of lake drainages and the influence on the front of Zachariae glacier. We have also discussed the Neckel et al. (2020) study which looks into drainage in our region. Specifically, line 425-431 and the supplement are changes for this comment.

5) I think a few figures need substantial work to make the study clearer- Fig 3 and 1 in particular.
We have now changed the vast majority of the figures based on both reviewer comments. We answer the more specific comments in the section below.

6) There are parts where the structure and paragraphs needs some review. I ask the authors to review this generally- assess whether material is in the correct section (i.e intro v results v discussion v conclusion), and check for sentences that seem out of place with theme of paragraph. Some key suggested changes:

- The NAO is not mentioned at all in the introduction and then takes substantial space in the discussion. I think this needs to be indicated as a theme in the intro given its significance- some of its discussion in the discussion section can be put in the intro.
  Thank you for your suggestion. We agree that this should have been introduced in the introduction. We have now moved the first NAO/teleconnections paragraph from the discussion to the introduction.

  A large part of the conclusion seems out of place. New discussion is brought in- e.g the Neckel et al. reference part.
  We have now moved the Neckel section to the discussion and altered other parts of the conclusion to strengthen it, see answer to comment below.

7) I think the conclusion needs substantial work. In addition to the above comment, in addition to the summaries of each year, I think the key takeaways need to be highlighted and importantly, there should be some discussion of the wider implications of the findings.
We have now changed the conclusion significantly to include key takeaways as well as wider implications of our study. We also highlighted areas of potential future work.

8) A small suggestion, perhaps it would be worth looking at the role of atmospheric rivers. See Mattingly et al. (2020).
The first author of the paper is currently working with a number of collaborators (including the author of the suggested paper you referenced) to look at the role of atmospheric rivers on the whole surface mass balance and melt production in this region. As this is quite a substantial amount of work and results, we will not include it in this manuscript. However, we have now included some reference to atmospheric rivers in the introduction.

Overall, I very much enjoyed reading this valuable study and hope that these points can help the authors to make the most out of the hard work they have put in. I provide line-by-line remarks/suggestions below.

Thank you so much for your positive comments. We hope that the changes we make throughout the manuscript, based on your and the other reviewer's suggestions, can improve the manuscript and be enjoyed by other members of the community.

Line-by-line and figure comments

Ln 34- "meltwater drainage channels"?
We have changed pattern to channels.

Ln 36- "and therefore they absorb…"
Changed, thank you.

Ln 47- "have investigated the relationship between the seasonal evolution…"
Changed, thank you.

Ln 55- Sundal et al. say "Since the ASTER image used for the test was acquired at a later stage in the melt season, the percentage of unidentified lake area at the start of the summer is likely to be somewhat higher than 12%"
This section now reads as: *'Sundal et al. (2009) used MODIS data to assess the lake area between 2003 and 2007 for 79°N amongst other locations. However, as the ASTER images were acquired at a later stage in the melt season, the percentage of unidentified lake area at the start of the summer is likely to be higher than 12% (Sundal et al. 2009).'*

Ln 56-58- this sentence about Sentinel 1 SAR seems out of place and without explanation, since the next sentence is instead about sentinel 2.
In this section, we wanted to highlight the limited amount of studies in this region, which includes a study using Sentinel 1. As it was unnecessary information about the satellite and made it seem out of place, we have now changed this section to: '*Winter estimates of liquid water area on the 79°N Glacier are also now available from Schröder et al. (2020).*'

L60-61- this sentence about what you do in this study does not fit well with the structure of the intro. It led me to think that you were about to elaborate on your own approach, but you go back to background discussion.
Agreed- we have now removed this sentence.

L66 – please provide some elaboration on the point about teleconnection signals.
Thank you for the comment. Reviewer 1 also requested some more information here. Therefore, we have moved a section from the discussion to the introduction (see response to key point 6).

L79- "whether **and how**".
Changed. Thank you.

L93- it would be more useful to give the approximate return time at high latitudes than the equator.
This has been updated for this region.

L167-168 – Specify for where this fact refers to. Also, I suggest leading the paragraph/section with your results and mention this later- it seems odd to lead a results section with results from the literature.

We have now changed the opening sentence to remove the literature results and to specify the location. It reads as: '*Here, we highlight the important lake characteristics and analyse the climatic and topographic controls responsible for the spatial and temporal distribution of SGLs on 79 °N Glacier, as detected by Hochreuther et al. (2021) from 2016 to 2019.*'

L175-180- I note an inconsistent use of tense. I don't mind if in present or past tense, but check for consistency.

Thank you for pointing this out. We changed it to past tense.

L178 – Careful to be specific. Does July refer to July in all years? Have just been referring to June 2017 so not totally clear. Check always clear with this. Also consider starting a new paragraph here.

Here, we have re-structured so that the first paragraph describes the more general changes which occur each year. The second paragraph now explains the interannual differences more specifically.

L203/204- "largest/smallest peak **total** SGL area"- check clear on this sort of thing in all instances.

Thank you, we have changed the two specific sentences you suggested and another now at line 254.

L207-209- Rephrase for clarity. I think part of the issue is 'close up' can be read as in 'close up to something'.

Close-up has been changed to '*a period when the SGLs freeze up*'.

L215-216- Rephrase- something like "Given sufficient meltwater availability, the location of lake formation…"

Changed to how you suggested. Thank you.

L223-224- You say earlier that you don't identify lakes below the grounding line due to issues with the lack of DEM and them moving. So I assume this is based on visual analysis? If so, mention in the methods that you will do this.

We have now included this in the method section 2.1: '*Description of the SGLs on the floating tongue throughout the paper reflect only visual inspection of the satellite images.*'

L223- "the lakes move position laterally" is not clear. Do you mean that they advect down glacier with the flow of the glacier (e.g as in Macdonald et al. 2018 and Langley et al. 2016)?

Yes, they move down glacier. The satellite images have similar linear features as those in the Macdonald et al. 2018 paper. We have added to this sentence. It now reads as: '*Below the grounding line of 79°N Glacier (on the floating tongue), the lakes advect downstream with the flow of the glacier towards the ocean (not shown), in a similar fashion to those observed on Petermann Glacier (Macdonald et al. 2018).*'

L225-226- please rephrase this for clarity

This now reads as: The SGL area in 2016 and 2019 is larger compared to 2017 and 2018. This interannual change in SGL area is due to the inland expansion of lakes to higher elevations (Figure 3), as opposed to the development of new lakes at lower elevations.

L233-235- it seems odd to make a first mention of the depth here. If you make some assessment of depth in this way, you should mention it in the methods section. Also, im uncertain what 'not shown' means- I think not in any of the figures? It would be good to at least show an example in the supplement figure. If you need to convey 'not shown' anywhere, please state what you mean more specifically.
What we want to express here is a comparative relation, not a real assessment (or even measure) of depth. We added an explanation to the methods part why depth was not calculated or estimated. Additionally, we added a figure to the supplement to visualize the difference in blue spectrum saturation between the low- and high elevation lakes (Supplement Fig.1)

L236-238- this paragraph seems out of place.
We have now moved part of this sentence to the first paragraph of the section '*However, above the grounding line, lakes develop in the same depression or location each year'*. The second sentence remains but has been moved to the start of section 3.3 to introduce the climate controls section.

Ln 246 – I think for this audience you should explain what 'skin temperature' means.
Done. Thank you.

L270 - "agreement **with"**
Changed, thank you.

L430-432 – please be explicit about what you point is here. I assume you're suggesting that the calving event possibly implies warmer temperatures.
We don't assume that the calving event *alone* implies warmer temperatures, but the calving event combined with the extensive melt pond formation and thin/broken sea ice does imply warmer temperatures.  The sentence now reads as: '*However, satellite images reveal extensive surface melt pond formation, very thin and broken sea ice, and a 50km2 calving event of Spalte Glacier was also recorded this year (Figure S1). When taken altogether, these characteristics point to particularly warm temperatures across the whole region*.

L469- Do you mean to say "Not only local meteorology… but also SMB prior to pond development". A reader might think from the sentence structure that you're implying "not only local but regional". Please state.
Yes, you are correct, we have changed this to: '*Therefore, not only the local meteorology but also the SMB controls the SGL development, especially at higher elevations.'*

L472-3 I believe here you mean across the Greenland ice sheet, but since you don't state it, it might read as if you mean the study area.
We have now included '*over the GIS'* to make it clear. Thank you.

L474- I would lead with comparison to Greenland, see main points at the start.

We have now included Greenland studies. Please see manuscript and answer to main points above.

L493-4- "however **a high amount of** accumulation can… and **reduce** melting"
Changed. Thank you.

L505- "second largest **total** (?) SGL area". And I think I know what you mean by 'spread' (a broader area) but please explain.
That's right. We have changed it to: '*Summer 2016 saw the second largest total SGL area and spatial distribution of SGLs.*'

L555- "While we do not consider melt water cannels in our analysis and focus only on SGLs..."
Changed, thank you.

L558-61- state where this Schröder study focuses on. And I do not think you need to explain here that SI SAR works without sunlight.
Added 79N Glacier and removed the sentence about SAR.

L564- Given the title of the paper refers to 79N glacier, it seems odd to summarize the study as being of the 'north east Greenland Ice stream'.
Yes we agree. We have now written 79°N Glacier (northeast Greenland).

L573-75- leave out the reference to Schröder in conclusion, leave that to earlier sections and just make your assessment here.
Changed as suggested.

L580-81- This sentence about volume measurements is out of place in the paragraph and section.
We have now moved this section to the discussion and altered the conclusion. See main points above.

Figures:
Figure 1: please provide a small map showing location within Greenland. (Newly released Qgreenland may help with this). Please mark on the grounding line. Please label to NE Greenland Ice Stream. This is mentioned in the paper but is not clear how it fits with 79N glacier. I suggest showing and labelling the wider study area in a satellite image/mosaic. Then show terrain height on that or make it a separate panel. Is it important to show calving in main figure? The calving event is a very minor part of the paper.
Thank you for your suggestions. We have now included an insert of Greenland and the NEGIS (with a label). We decided to use ice velocity data to differentiate NEGIS from the ice sheet, as this is the most common delineator. We have now labelled the grounding line and included a mosaic of the satellite granules. The calving image has been moved to the supplement.

Figure 2:Since you refer to months in the text, please put them on the x axis. I find this color scale difficult- the change from 0-300 isn't a lot, making it hard to desciper medium or smaller changes.

Regarding the colour scale, we changed it to a binary (blue vs. grey) scale, to display the direction of change (positive/negative) and not duplicate information (colour scale and bar height). The magnitude of change can now be read from both the left and right y-axis (same scale). The figure now also includes the total lake area as line graph. However, we did not add months to the axis, so that it is in line with figure 5, and we added day of the year to the text.

Figure 3: I find it very difficult to make an assessment of what is happening in this figure without spending a while looking at it. Perhaps consider having a zoomed in sample site. If you use this approach please label July/August on the figures themselves, and consider using a satellite image as the background (although I understand that may not work if other surface features make it busy).

We have now split this figure into 4 different years, so that the interannual differences can be easily seen. Furthermore, the two colours now show the lake distribution at peak total SGL area and the period of freeze-up. We did not put a satellite image as the background as it added confusion to the figure.

Figure 5: Can you make the scale consistent with Fig2? If not the scale itself, please make the labels consistent (I.e add day of year and add months to label on Figure 2).

Thank you for your suggestion. Based on this and the other reviewer's comments, we have changed Figure 5 to 4 panels, split by year, with additional information from Table 2. The day of the year has been used for the label and the axis is now consistent with figure 2.

---

## Author Response (AR2)

Dear Reviewers and Editor Lhermitte,

Thank you very much for the fast response and positive feedback on our manuscript and the verdict of acceptance after minor corrections. We are pleased to upload the final version of the manuscript and a tracked-changes version. We also outline our response to reviewer 1 below. We very much appreciate the time taken for a second review and the additional comments. We are pleased with the manuscript now and are happy to have it published in The Cryosphere.

Best wishes,
Jenny Turton, on behalf of all authors.

Summary

Turton et al. have submitted. A thorough revision to the comments of reviewer #2 and myself. The manuscript is a pleasure to read and the new content helps to clearly visualise and communicate the valuable contributions of the study. I have very few further comments (below).

Comments

L62 and elsewhere – I still think the authors should avoid referring to Sentinel-2 as VHR – although it may be in the modelling world, VHR has a very definite meaning when applied to satellites.
Thank you for your suggestion, we have now removed this from the manuscript throughout.

It is good to see the teleconnection patterns now represented in the introduction. However, I think it could be slightly better integrated. For instance, sentence L76-L79 refers to 'these stand-out years' (L77), which I think, after checking the track changes document, is a reference to 2007, 2010 and 2012 (L64), with the new content inserted in between. A rewrite should better integrate the text. I also think it is worth a sentence at the start of the paragraph highlighting the (complete?) lack of studies linking teleconnections to SGLs. This will better set the stage for the study.
Thank you for catching this. We have now re-written part of this paragraph for better understanding and to integrate it better with the rest of the introduction. More specifically, we have changed 'these stand-out years' to 'extreme melting years' and included an additional sentence about the extreme melting years. We have also included a sentence about the lack of teleconnection-melt pond studies, however at the end of the paragraph, to better connect with the following section. This is written in answer to the next comment. Or see lines 64 to 80 for changes.

L74 – "More recently, atmospheric rivers…" Unclear why paragraph begins here. Important to include but in above paragraph on atmospheric events more generally maybe.
We have now added an additional line to the start of this paragraph and one at the end of the previous paragraph to connect the two and enable a better flow between large-scale and smaller-scale atmospheric features. Specifically: *'Extreme Greenland-wide melt seasons, such as in 2012, have been linked to specific teleconnection patterns (Tedesco et al. 2013),*

*however no studies have assessed the potential role of teleconnections on the development of SGLs or localised melt conditions.*
*Along with large-scale teleconnection influences, smaller-scale mesoscale processes also influence the climate and melting over Greenland. Recently, atmospheric rivers…'*

Section 3.3.5, beginning L471 – this new text is incredibly useful, but I wonder whether it is interpretation that belongs in the discussion (perhaps as an opener) rather than in results.
Thank you for your positive comments. We have now moved this section to the discussion as an introductory paragraph. We have also included an additional sentence at the start and end to better flow between paragraphs. Please see lines 541 onwards (for tracked changed manuscript or lines 511 in non-tracked version).

Figure 1: Commendable new panels but text should now be larger to be readable.
Our apologies, we have now made all the text larger.

Figure 6 and 7 caption: Perhaps 'monthly average' wind direction and speed?
This has been changed as suggested, thank you.